# TRAINING ADVERSARIALLY ROBUST SNNS WITH GRADIENT SPARSITY REGULARIZATION

## ABSTRACT

Spiking Neural Networks (SNNs) have attracted much attention for their energy-efficient operations and biologically inspired structures, offering potential advantages over Artificial Neural Networks (ANNs) in terms of interpretability and energy efficiency. However, similar to ANNs, the robustness of SNNs remains a challenge, especially when facing adversarial attacks. Existing techniques, whether adapted from ANNs or specifically designed for SNNs, have shown limitations in training SNNs or defending against strong attacks. In this paper, we present a novel approach to enhance the robustness of SNNs through gradient sparsity regularization. We observe that SNNs exhibit greater resilience to random perturbations compared to adversarial perturbations, even at larger scales. Motivated by this finding, we aim to minimize the gap between SNNs under adversarial and random perturbations, thereby improving their overall robustness. To achieve this, we theoretically prove that this performance gap is upper bounded by the gradient sparsity of the output probability after the softmax layer with respect to the input image, laying the groundwork for a practical strategy to train robust SNNs by regularizing the gradient sparsity. The effectiveness of our approach is validated through extensive experiments conducted on the CIFAR-10 and CIFAR-100 datasets. The results demonstrate enhancements in the robustness of SNNs. Overall, our work contributes to the understanding and improvement of SNN robustness, highlighting the importance of considering gradient sparsity in SNNs.

## 1 INTRODUCTION

Although Artificial Neural Networks (ANNs) have achieved impressive performance across various tasks (He et al., 2016; Mishra et al., 2021; Khan et al., 2023), they are often plagued by complex computations and limited interpretability (Liu et al., 2021; Lipton, 2018). In recent years, Spiking Neural Networks (SNNs) have garnered significant attention in the field of artificial intelligence due to their energy-efficient operations and biologically-inspired architectures (Maass, 1997; Zenke et al., 2021). In SNNs, neurons simulate changes in membrane potentials and transmit information through spike trains (Roy et al., 2019). These characteristics allow for a certain level of biological interpretation while avoiding the extensive and complex matrix multiplication operations inherent in ANNs. Remarkably, SNNs have achieved competitive performance with ANNs on various classification datasets (Sengupta et al., 2019; Deng et al., 2022; Fang et al., 2021a).

Similar to ANNs (Tanay & Griffin, 2016; Stutz et al., 2019; Liu et al., 2022), the issue of robustness poses a significant challenge for SNNs (Sharmin et al., 2019; 2020; Kundu et al., 2021). When subjected to imperceptible perturbations added to input images, SNNs can exhibit misclassifications, which are known as adversarial attacks. Developing techniques to train SNNs for adversarial robustness remains an ongoing problem in the research community. Some successful techniques originally designed for ANNs have been adapted for use with SNNs, including adversarial training (Madry et al., 2018; Ho et al., 2022; Ding et al., 2022) and certified training(Zhang et al., 2020; 2021; Liang et al., 2022). Additionally, there are SNN-specific methods proposed to improve robustness, such as temporal penalty configurations (Leontev et al., 2021), and specialized coding schemes (Sharmin et al., 2020). However, these methods either prove challenging to train on SNNs (Liang et al., 2022) or exhibit limited effectiveness against strong attacks (Sharmin et al., 2020).

In this paper, we present a novel approach to enhance the robustness of SNNs by considering the gradient sparsity with respect to the input image. We find that SNNs exhibit greater robustness to random perturbations, even at larger scales, compared to adversarial perturbations. Building upon this observation, we propose to minimize the performance gap between an SNN subjected to adversarial perturbations and random perturbations, thereby enhancing its overall robustness. The main contributions of our work are as follows.

- We analyze the robustness of SNNs and reveal that SNNs exhibit robustness against random perturbations even at significant scales, but display vulnerability to small-scale adversarial perturbations.
- We provide theoretical proof that the gap between the robustness of SNNs under these two types of perturbations is upper bounded by the sparsity of gradients of the output probability with respect to the input image.
- We propose to incorporate gradient sparsity regularization into the loss function during SNN training to narrow the gap, thereby boosting the robustness of SNNs.
- Extensive experiments on the CIFAR-10 and CIFAR-100 datasets validate the effectiveness of our method, which significantly improves the robustness of SNNs.

## 2 RELATED WORK

### 2.1 LEARNING ALGORITHMS OF SNNS

The primary objective of most SNN learning algorithms is to achieve high-performance SNNs with low latency. Currently, the most effective and popular learning algorithms for SNNs are the ANN-SNN conversion (Cao et al., 2015) and supervised learning (Wu et al., 2018). The ANN-SNN conversion method aims to obtain SNN weights from pre-trained ANNs with the same network structure. By utilizing weight scaling, threshold balancing, and quantization training techniques, well-designed ANN-SNN algorithms can achieve lossless performance compared to the original ANN (Diehl et al., 2015; Sengupta et al., 2019; Han et al., 2020; Deng & Gu, 2021; Ho & Chang, 2021; Li et al., 2021; Bu et al., 2022; Hu et al., 2023). However, the converted SNNs often require larger timesteps to achieve high performance, resulting in increased energy consumption. Moreover, they lose temporal information and struggle to process neuromorphic datasets. The supervised learning approach directly employs the backpropagation algorithm to train SNNs with fixed timesteps. Wu et al. (2018; 2019) borrowed the idea from the Back Propagation Through Time (BPTT) in RNN learning and proposed the Spatio-Temporal-Back-Propagation (STBP) algorithm. They approximate the gradient of spiking neurons using surrogate functions (Neftci et al., 2019). While supervised training significantly improves the performance of SNNs on classification tasks (Kim et al., 2020; Kim & Panda, 2020; Lee et al., 2020; Fang et al., 2021b; Zheng et al., 2021; Deng et al., 2022; Guo et al., 2022; Yao et al., 2022; Mostafa, 2017; Bohte et al., 2000; Zhang & Li, 2020; Zhang et al., 2022; Zhu et al., 2022), SNNs still fall behind ANNs in terms of generalization and flexibility. Challenges such as gradient explosion/vanishing and spike degradation persist in SNNs.

### 2.2 DEFENSE METHODS OF SNNS

Methods for improving the robustness of SNNs can be broadly categorized into two classes. The first class draws inspiration from ANNs. A typical representative is adversarial training, which augments the training set with adversarial examples generated by attacks (Madry et al., 2018; Tramèr et al., 2018; Wong et al., 2020). This approach has been shown to effectively defend against attacks that are used in the training phase. Another method is certified training, which utilizes certified defense methods to train a network (Wong & Kolter, 2018; Xu et al., 2020). Certified training has demonstrated promising improvements in the robustness of ANNs (Zhang et al., 2020), but its application to SNNs remains challenging (Liang et al., 2022). The second category consists of SNN-specific techniques designed to enhance robustness. On one hand, the choice of encoding the continuous intensity of an image into 0-1 spikes can impact the robustness of SNNs. Recent studies have highlighted the Poisson encoder as a more robust option (Sharmin et al., 2020; Kim et al., 2022). However, the Poisson encoder generally yields worse accuracy on clean images than the direct encoding, and the robustness improvement caused by the Poisson encoder varies with the number of timesteps

used. On the other hand, researchers have recognized the unique temporal dimension of SNNs and developed strategies related to temporal aspects to improve robustness Nomura et al. (2022). Apart from the studies on static datasets, there are works that attempt to perform adversarial attacks and defenses on the DVS (Dynamic Vision Sensors) dataset (Marchisio et al., 2021b). In this paper, we mainly focus on the direct encoding of input images. We propose a gradient sparsity regularization strategy to improve SNNs' robustness with theoretical guarantees. Moreover, this strategy can be combined with adversarial training to further boost the robustness of SNNs.

# 3 PRELIMINARY

## 3.1 NEURON DYNAMICS IN SNNs

Similar to previous works (Wu et al., 2018; Sharmin et al., 2020; Rathi & Roy, 2021), we consider the commonly used Leaky Integrate-and-Fire (LIF) neuron model due to its efficiency and simplicity, the dynamic of which can be formulated as follows:

$$u_i^l[t] = \tau u_i^l[t-1](1 - s_i^l[t-1]) + \sum_j \boldsymbol{w}_{i,j}^{l-1} \boldsymbol{s}_j^{l-1}[t], \tag{1}$$

$$s_i^l[t] = H(u_i^l[t] - \theta). \tag{2}$$

Equation(1) describes the membrane potential of the $i$-th neuron in layer $l$, which receives the synaptic current from the $j$-th neuron in layer $l-1$. Here $\tau$ represents the membrane time constant of the neuron and $t$ denotes the discrete time step ranging from 1 to $T$. The variable $u_i^l[t]$ represents the membrane potential of the $i$-th neuron in layer $l$ at the time step $t$. $w_{i,j}$ denotes the synaptic weight between the two neurons, and $\boldsymbol{s}_j^{l-1}[t]$ represents the binary output spike of neuron $j$ in layer $l-1$. For simplicity, the resting potential is assumed to be zero so that the membrane potential will be reset to zero after firing. Equation(2) defines the neuron fire function. At each time step $t$, a spike will be emitted when the membrane potential $u_j^l[t]$ surpasses a specific threshold $\theta$. The function $H(\cdot)$ corresponds to the Heaviside step function, which equals 0 for negative input and 1 for others.

## 3.2 ADVERSARIAL ATTACKS FOR SNNs

Preliminary explorations have revealed that SNNs are also susceptible to adversarial attacks (Sharmin et al., 2020; Kundu et al., 2021; Liang et al., 2021; Marchisio et al., 2021a; Ding et al., 2022). Well-established techniques such as the Fast Gradient Sign Method (FGSM) (Goodfellow et al., 2015) and Projected Gradient Descent (PGD) (Madry et al., 2018) can generate strong adversaries that threaten SNNs.

**FGSM** is a straightforward none-iterative attack method, expressed as follows:

$$\hat{\boldsymbol{x}} = \boldsymbol{x} + \epsilon \, \text{sign}(\nabla_{\boldsymbol{x}} \mathcal{L}(f(\boldsymbol{x}), y)), \tag{3}$$

where $\boldsymbol{x}$ and $\hat{\boldsymbol{x}}$ represent the original image and the adversarial example respectively, $\epsilon$ denotes the perturbation bound, $\mathcal{L}$ refers to the loss function, $f(\cdot)$ represents the neural network function, and $y$ denotes the label data.

**PGD** is an iterative extension of FGSM, which can be described as follows:

$$\hat{\boldsymbol{x}}^k = \Pi_\epsilon \{\boldsymbol{x}^{k-1} + \alpha \, \text{sign}(\nabla_{\boldsymbol{x}} \mathcal{L}(f(\boldsymbol{x}^{k-1}), y))\}, \tag{4}$$

where $k$ is the total iteration step and $\alpha$ is the step size. The operator $\Pi_\epsilon$ projects the adversarial examples onto the space of the $\epsilon$ neighborhood in the $\ell_\infty$ norm around $\boldsymbol{x}$.

# 4 METHODOLOGY

In this paper, we first compare the vulnerability of SNNs to random perturbations versus adversarial perturbations. We highlight that SNNs exhibit significant robustness against random perturbations, whereas they are more susceptible to adversarial perturbations. Then we quantify the disparity between adversarial vulnerability and random vulnerability and prove that the disparity is upper bounded by the gradient sparsity of the output probability after the final softmax layer with respect to the input image. Based on this, we propose a novel approach to enhance the robustness of SNNs by introducing sparsity regularization of gradients in the training phase and incorporating this regularization into the learning rule of SNNs.

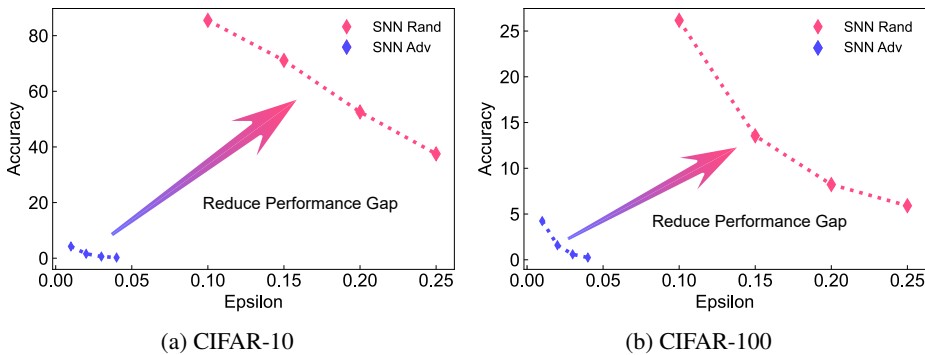

(a) CIFAR-10              (b) CIFAR-100

Figure 1: Comparison of the Random Vulnerability and Adversarial Vulnerability of SNNs

### 4.1 COMPARE RANDOM VULNERABILITY AND ADVERSARIAL VULNERABILITY

The objective of adversarial attacks is to deliberately alter the output vector $f(\boldsymbol{x})$ in order to change the classification result. Since the classification result is decided by $\arg\max_i f_i$, attackers aim to substantially decrease the value of the component corresponding to the true label of $\boldsymbol{x}$. For simplicity, we suppose $\boldsymbol{x}$ with label $y$ belonging to the $y$-th class, so that the value of $f_y(\boldsymbol{x})$ affects the classification result significantly. Therefore, the stability of $f_y(\boldsymbol{x})$ becomes crucial for SNN robustness, particularly the value of $f_y(\hat{\boldsymbol{x}}) - f_y(\boldsymbol{x})$ when subjected to small perturbations on $\boldsymbol{x}$.

To initiate our analysis, we define the random and adversarial vulnerability of an SNN, denoted as $f$, at a specific point $\boldsymbol{x}$ under an $\ell_p$ attack of size $\epsilon$.

**Definition 1.** (Random Vulnerability) The random vulnerability of $f$ at point $\boldsymbol{x}$ to an $\ell_p$ attack of size $\epsilon$ is defined as the expected value of $(f_y(\boldsymbol{x} + \epsilon \cdot \delta) - f_y(\boldsymbol{x}))^2$, where $\delta$ follows a uniform distribution within the unit $\ell_p$ ball, and $y$ represents the class of $\boldsymbol{x}$ belonging to. Mathematically, it can be expressed as:

$$\rho_{\mathrm{rand}}(f, \boldsymbol{x}, \epsilon, \ell_p) = \mathbb{E}_{\delta \sim U\{\|\delta\|_p \leqslant 1\}} \left( f_y(\boldsymbol{x} + \epsilon \cdot \delta) - f_y(\boldsymbol{x}) \right)^2. \tag{5}$$

**Definition 2.** (Adversarial Vulnerability) The adversarial vulnerability of $f$ at point $\boldsymbol{x}$ to an $\ell_p$ attack of size $\epsilon$ is defined as the supremum of $(f_y(\boldsymbol{x} + \epsilon \cdot \delta) - f_y(\boldsymbol{x}))^2$, where $\delta$ follows a uniform distribution within the unit $\ell_p$ ball, and $y$ represents the class of $\boldsymbol{x}$ belonging to. Mathematically, it can be expressed as:

$$\rho_{\mathrm{adv}}(f, \boldsymbol{x}, \epsilon, \ell_p) = \sup_{\delta \sim U\{\|\delta\|_p \leqslant 1\}} \left( f_y(\boldsymbol{x} + \epsilon \cdot \delta) - f_y(\boldsymbol{x}) \right)^2. \tag{6}$$

To gain a deeper understanding of the disparity in vulnerability between random perturbations and adversarial perturbations in SNNs, we conduct a small-scale experiment with a primary emphasis on $\ell_\infty$ attacks. The choice to prioritize $\ell_\infty$ attacks over other $\ell_p$ attacks is justified due to the widespread use of $\ell_\infty$ constraints in various attack methods. Adversarial examples generated under $\ell_\infty$ attacks tend to be more destructive compared to those generated under $\ell_0$ and $\ell_2$ attacks. Therefore, focusing on $\ell_\infty$ attacks allows us to assess the SNN's robustness against the most severe random and adversarial perturbations.

In our experiment, we evaluate the performance of a well-trained SNN $f$ with the VGG-11 architecture. Our primary objective is to examine the impact of both random and adversarial perturbations on the SNN's classification results. For a given image $\boldsymbol{x}$, we added random perturbations uniformly drawn from a hyper-cube $\{\delta_{\mathrm{rand}} : \|\delta_{\mathrm{rand}}\|_\infty \leqslant \epsilon\}$ to the original image, resulting in the perturbed image $\hat{\boldsymbol{x}}_{\mathrm{rand}}$. Additionally, we employed a $\epsilon$-sized FGSM attack to generate an adversarial example $\hat{\boldsymbol{x}}_{\mathrm{adv}}$ from $\boldsymbol{x}$. Subsequently, we individually input both $\hat{\boldsymbol{x}}_{\mathrm{rand}}$ and $\hat{\boldsymbol{x}}_{\mathrm{adv}}$ into the network $f$ to observe any changes in the classification results. We evaluate the classification accuracy of the perturbed images on the test set of CIFAR-10/CIFAR-100 under both random and adversarial perturbations, respectively. The results, as depicted in Figure 1, yield the following key findings:

**Observation 1.** *SNNs exhibit robustness against random perturbations even when the perturbation scale is significant, but display vulnerability to small-scale adversarial perturbations.*

## 4.2 QUANTIFY THE GAP BETWEEN RANDOM VULNERABILITY AND ADVERSARIAL VULNERABILITY

The aforementioned observations indicate that SNNs demonstrate notable robustness to random perturbations in comparison to adversarial perturbations. To enhance the robustness of SNNs against adversarial attacks, a natural approach is to minimize the disparity between adversarial vulnerability and random vulnerability.

To measure the disparity in vulnerability between the SNN $f$ under adversarial perturbations and random noise, we employ the ratio of $\rho_{\text{adv}}(f, \boldsymbol{x}, \epsilon, \ell_\infty)$ and $\rho_{\text{rand}}(f, \boldsymbol{x}, \epsilon, \ell_\infty)$. We make the assumption that $\rho_{\text{rand}}(f, \boldsymbol{x}, \epsilon, \ell_\infty) \neq 0$, indicating that $f$ is not a constant function. This assumption aligns with the practical reality of SNNs in real-world applications. Optimizing the ratio of $\rho_{\text{adv}}(f, \boldsymbol{x}, \epsilon, \ell_\infty)$ and $\rho_{\text{rand}}(f, \boldsymbol{x}, \epsilon, \ell_\infty)$ directly is a challenging task. However, we are fortunate to present a mathematical proof that establishes an upper bound for this ratio based on the sparsity of $\nabla_{\boldsymbol{x}} f_y$. Specifically, we have the following theorem.

**Theorem 1.** *Suppose $f$ is a differentiable SNN by surrogate gradients, and $\epsilon$ represents the magnitude of an attack, assumed to be small enough. Given an input image $\boldsymbol{x}$ with corresponding label $y$, the ratio of $\rho_{adv}(f, \boldsymbol{x}, \epsilon, \ell_\infty)$ and $\rho_{rand}(f, \boldsymbol{x}, \epsilon, \ell_\infty)$ is upper bounded by the sparsity of $\nabla_{\boldsymbol{x}} f_y$:*

$$3 \leqslant \frac{\rho_{adv}(f, \boldsymbol{x}, \epsilon, \ell_\infty)}{\rho_{rand}(f, \boldsymbol{x}, \epsilon, \ell_\infty)} \leqslant 3\|\nabla_x f_y(\boldsymbol{x})\|_0. \tag{7}$$

The proof is provided in Appendix A.

This theorem illustrates that the disparity between the adversarial vulnerability and random vulnerability is upper bounded by the sparsity of $\nabla_{\boldsymbol{x}} f_y$. It provides valuable insights into the correlation between gradient sparsity and the disparity in robustness exhibited by SNNs when subjected to different perturbations with $\ell_\infty$ attacks. According to Theorem 1, we can infer that a sparser gradient contributes to closing the robustness gap between SNN $f$ under worst-case scenarios and its robustness under random perturbations.

From an intuitive perspective, minimizing the $\ell_0$ norm of $\nabla_{\boldsymbol{x}} f_y(\boldsymbol{x})$ serves to bring $\boldsymbol{x}$ closer to a local maximum point. In an ideal scenario, this would entail trapping $\boldsymbol{x}$ within a local maximum, effectively rendering attackers unable to generate adversarial examples through gradient-based methods. By introducing the sparsity constraint for each $\boldsymbol{x}$ in the training set, we encourage to learn an SNN $f$ where input images tend to remain close to extreme points or trapped in local maximums. This makes it challenging to perturb $f_y(\boldsymbol{x})$ with small perturbations, thereby enhancing the robustness of the SNN $f$.

## 4.3 LOSS FUNCTION WITH SPARSITY REGULARIZATION

To promote sparsity of the gradients, a straightforward approach is to incorporate the $\|\nabla_{\boldsymbol{x}} f_y(\boldsymbol{x})\|_0$ term into the training loss. Specifically, the training loss for an input image $\boldsymbol{x}$ can be defined as:

$$\mathcal{L}(\boldsymbol{x}, y, w) = \text{CE}(f(\boldsymbol{x}, w), y) + \lambda \|\nabla_{\boldsymbol{x}} f_y(\boldsymbol{x}, w)\|_0, \tag{8}$$

where $w$ represents the weight parameters of $f$, $\text{CE}(\cdot)$ is the cross-entropy loss, and $\lambda$ denotes the coefficient parameter controlling the strength of the sparsity regularization.

Here we give the formulation of $\nabla_{\boldsymbol{x}} f_y(\boldsymbol{x})$ and the detailed derivation is provided in Appendix J. We consider an SNN $f$ with a final layer denoted as $L$. The total number of neurons in layer $L$ is denoted by $N$, and the time-step $t$ ranges from 1 to $T$. The output vector depends on the collective outputs across all time-steps. For a given input image $\boldsymbol{x}$, the $i$-th component of $f(\boldsymbol{x}) \in \mathbb{R}^n$ is:

$$f_i = \text{softmax}_i \left( \sum_{t=1}^{T} s_1^L(t), \ldots, \sum_{t=1}^{T} s_N^L(t) \right). \tag{9}$$

---

**Algorithm 1** Training Algorithm

---

**Input**: Spiking neural network $f(\boldsymbol{x}, w)$ with parameter $w$  Learning rate $\eta$; Step size $h$ of finite differences
**Output**: Regularize trained parameter $w$

1: **for** epoch=0 **to** n **do**
2:     Sample minibatch $\{(\boldsymbol{x}^i, y^i)\}_{i=1,\dots,m}$ from Dataset $\mathcal{D}$
3:     **for** i=0 **to** m **do**
4:         $g^i = \nabla_x \text{softmax}_{y^i}(f(\boldsymbol{x}^i, w))$
5:         $d^i = \text{sign}(g^i) \leftarrow$ the difference direction
6:         $\hat{\boldsymbol{x}}^i = \boldsymbol{x}^i + h d^i$
7:         $\mathcal{L}(\boldsymbol{x}^i, y^i, w) = \text{CE}(f(\boldsymbol{x}^i), y^i) + \frac{\lambda}{h} \left\| (\text{softmax}_{y^i}(f(\boldsymbol{x}^i, w)) - \text{softmax}_{y^i}(f(\hat{\boldsymbol{x}}^i, w))) \right\|$
8:         $w \leftarrow w - \eta \nabla_w \mathcal{L}(\boldsymbol{x}^i, y^i, w)$
9:     **end for**
10: **end for**

---

Given an input image $\boldsymbol{x}$ belonging to the $y$-th class, $\nabla_{\boldsymbol{x}} f_y(\boldsymbol{x}, w)$ can be formulated as:

$$\nabla_{\boldsymbol{x}} f_y(\boldsymbol{x}, w) = \sum_{i=1}^{N} \left( \frac{\partial \text{softmax}_y}{\partial \sum_t s_i^L[t]} \left( \sum_{t=1}^{T} \sum_{\tilde{t}=1}^{t} \nabla_{\boldsymbol{x}[\tilde{t}]} s_i^L[t] \right) \right). \tag{10}$$

It is worth noting that the optimization problem involving the $\ell_0$ norm is known to be NP-hard (Natarajan, 1995). To circumvent this challenge, we employ the $\ell_1$ norm as a substitute for the $\ell_0$ norm because the $\ell_1$ norm serves as a convex approximation to the $\ell_0$ norm (Ramirez et al., 2013). Additionally, to avoid the computational burden associated with calculating the back-propagation of $\|\nabla_{\boldsymbol{x}} f_y(\boldsymbol{x}, w)\|_1$, we adopt a finite difference approximation for this term (Finlay & Oberman, 2021). In our case, we approximate the gradient regularization term using the following finite differences:

**Proposition 1.** *Let $d$ denote the signed input gradient direction: $d = sign(\nabla_{\boldsymbol{x}} f_y(\boldsymbol{x}, w))$. Let $h$ be the finite difference step size. Then, the $\ell_1$ gradient norm can be approximated as follows:*

$$\|\nabla_{\boldsymbol{x}} f_y(\boldsymbol{x}, w)\|_1 \approx \left| \frac{f_y(\boldsymbol{x} + h \cdot d, w) - f_y(\boldsymbol{x}, w)}{h} \right| \tag{11}$$

The proof is provided in the Appendix B. Finally, the training loss (Equation (8)) is rewritten as:

$$\mathcal{L}(\boldsymbol{x}, y, w) = \text{CE}(f(\boldsymbol{x}, w), y) + \lambda \left| \frac{f_y(\boldsymbol{x} + h \cdot d, w) - f_y(\boldsymbol{x}, w)}{h} \right|. \tag{12}$$

The overall training algorithm is presented as Algorithm 1.

**Differences with the related work.** We compare the proposed sparsity regularization with the Gradient Regularization (GR) proposed by Finlay & Oberman (2021). While GR's regularization term depends on the choice of the classification loss function (cross-entropy etc.), the sparsity regularization is solely tied to the model $f$ and remains constant across different loss functions. Additionally, GR aims to minimize the upper bound of the adversarial training loss, whereas our sparsity regularization is independent of adversarial training. More details are illustrated in the Appendix H.

## 5 EXPERIMENT

In this section, we evaluate the performance of our method on image classification tasks using the CIFAR-10/100 datasets. We adopt the experiment setting used in the previous work (Ding et al., 2022). Specifically, we use the VGG-11 architecture (Simonyan & Zisserman, 2014), WideResNet with a depth of 16 and width of 4 (WRN-16) (Zagoruyko & Komodakis, 2016). The timestep for the SNNs is set to 8. Throughout the paper, we use the IF neuron with a hard-reset mechanism as the spiking neuron. Further details regarding the training settings can be found in the Appendix C.

To generate adversarial examples, we employ different attacks, including FGSM (Goodfellow et al., 2015), PGD (Madry et al., 2018), and AutoPGD (Croce & Hein, 2020), with a fixed attack strength

Table 1: Comparison with the SOTA methods on classification accuracy (%) under attacks.

| Defense | Clean | White Box Attack | | | | Black Box Attack | | | |
|---|---|---|---|---|---|---|---|---|---|
| | | PGD10 | PGD30 | PGD50 | APGD10 | PGD10 | PGD30 | PGD50 | APGD10 |
| CIFAR-10    VGG-11 | | | | | | | | | |
| RAT | 90.44 | 11.53 | 7.08 | 6.41 | 3.26 | 43.29 | 40.17 | 40.16 | 47.50 |
| AT | 89.97 | 18.18 | 14.79 | 14.63 | 10.36 | 44.02 | 43.38 | 43.40 | 52.90 |
| SR* | 85.91 | 30.54 | 28.06 | 27.66 | 21.91 | 51.21 | 50.85 | 51.06 | 59.87 |
| CIFAR-10    WRN-16 | | | | | | | | | |
| RAT | 92.70 | 10.52 | 5.14 | 4.33 | 2.19 | 38.35 | 31.04 | 30.40 | 36.42 |
| AT | 90.97 | 17.88 | 14.89 | 14.62 | 9.13 | 43.99 | 42.09 | 41.55 | 52.10 |
| SR* | 85.63 | 39.18 | 37.04 | 36.74 | 29.03 | 50.84 | 50.23 | 49.97 | 57.93 |
| CIFAR-100    WRN-16 | | | | | | | | | |
| RAT | 69.10 | 5.72 | 3.58 | 3.26 | 2.08 | 22.61 | 18.77 | 18.26 | 25.23 |
| AT | 67.37 | 10.07 | 8.12 | 7.86 | 4.88 | 25.17 | 23.76 | 23.50 | 35.96 |
| SR* | 60.37 | 19.76 | 18.39 | 18.11 | 13.32 | 28.38 | 28.01 | 27.94 | 36.69 |

of $8/255$. For iterative attacks, the number of iterations is indicated in the attack name (e.g. PGD10). Since the choice of gradient approximation methods (Bu et al., 2023) and surrogate functions can affect the attack success rate (Xu et al., 2022), we consider an ensemble attack for SNNs. We utilize a diverse set of surrogate gradients and consider both STBP-based and RGA-based attacks. For each test sample, we conduct multiple attacks using all possible combinations of attack settings, and report the strongest attack. Robustness is evaluated in two scenarios: the white-box scenario, where attackers have knowledge of the target model, and the black-box scenario, where the target model is unknown to attackers. More detailed evaluation settings can be found in the Appendix D.

## 5.1 COMPARE WITH THE STATE-OF-THE-ART

We validate the effectiveness of our method by comparing it with the current state-of-the-art approaches, including Regularized Adversarial Training (RAT) (Ding et al., 2022) and Adversarial Training (AT) (Kundu et al., 2021). We use SR* to denote our sparsity regularization strategy with adversarial training. For RAT-SNN, we replicate the model following the settings outlined in the paper (Ding et al., 2022). As for all adversarial trained SNNs, we adopt a PGD5 attack with $\epsilon = 2/255$.

Table 1 reports the classification accuracy of the compared methods under ensemble attacks. Columns 3-6 highlight the substantial enhancement in SNN robustness achieved through our strategy in the white box scenario. Our proposed method consistently outperforms other state-of-the-art methods across all datasets and architectures. For instance, when subjected to 10-steps PGD attacks, VGG-11 trained with our strategy elevates classification accuracy from 11.53% (RAT) to an impressive 30.54% on CIFAR-10. Similarly, WRN-16 trained using our method exhibits a remarkable 15% boost in classification accuracy against PGD50 on CIFAR-100. In comparison to the Adversarial Training (AT) strategy, our method demonstrates a noteworthy enhancement in adversarial robustness, with a 10-20 percentage point improvement on both datasets under all attacks.

In contrast to white box attacks, all strategies exhibit better adversarial robustness against black box attacks (columns 7-10 in Table 1). When considering the CIFAR-10 dataset, models trained using any strategy achieve a classification accuracy of over 30% when subjected to PGD50. However, models trained with the SR* strategy consistently outperform other strategies in all scenarios. There exists a gap of 10%-20% in performance between models trained with RAT/AT and those trained with SR* on CIFAR-10, and a 5%-10% gap on CIFAR-100. These results clearly demonstrate the superiority of our approach over state-of-the-art methods.

## 5.2 ABLATION STUDY OF THE SPARSITY REGULARIZATION

We conduct additional experiments to validate the effectiveness of the sparsity regularization. In the ablation study, we compare the robustness performance of SNNs using different training strategies:

Table 2: Ablation study of the sparsity regularization.

| Dataset | Architecture | SR | AT | Clean | FGSM | RFGSM | PGD30 | PGD50 | APGD10 |
|---------|--------------|----|----|-------|------|-------|-------|-------|--------|
| CIFAR-10 | WRN-16 | ✗ | ✗ | 93.89 | 5.23 | 3.43 | 0.00 | 0.00 | 0.00 |
| | | ✗ | ✓ | 90.97 | 33.49 | 58.19 | 14.89 | 14.62 | 9.13 |
| | | ✓ | ✗ | 86.57 | 34.79 | 55.96 | 12.27 | 11.70 | 8.25 |
| | | ✓ | ✓ | 85.63 | 48.47 | 64.65 | 37.04 | 36.74 | 29.03 |
| CIFAR-100 | WRN-16 | ✗ | ✗ | 74.59 | 3.51 | 1.37 | 0.00 | 0.00 | 0.00 |
| | | ✗ | ✓ | 67.37 | 19.07 | 33.19 | 8.12 | 7.68 | 4.88 |
| | | ✓ | ✗ | 67.67 | 11.15 | 18.18 | 0.87 | 0.84 | 0.47 |
| | | ✓ | ✓ | 60.37 | 25.76 | 36.93 | 18.39 | 18.11 | 13.32 |

vanilla SNN, SR-SNN, AT-SNN, and SR-SNN with AT. The results on CIFAR-10 and CIFAR-100 are presented in Table 2, and the key findings are summarized as follows.

Firstly, it is crucial to note that vanilla SNNs exhibit poor adversarial robustness, with their classification accuracy dropping to a mere 5% when subjected to the FGSM attack. However, the SR strategy significantly enhances this performance, achieving a classification accuracy of 34.79% on the CIFAR-10 dataset and 11.15% on the CIFAR-100 dataset. Furthermore, combining the SR strategy with the AT strategy further boosts the robustness of SNNs, particularly against strong attacks like PGD50 and APGD10, resulting in a notable 10%-30% improvement.

Additionally, it is observed that the classification accuracy of robust SNNs on clean images may typically be slightly lower than that of baseline models. This phenomenon is consistent across all robustness training strategies. For example, WRN-16 models trained using any strategy exhibit a classification accuracy of less than 70% on clean images in CIFAR-100. Striking a balance between adversarial robustness and classification accuracy on clean images across all datasets remains an open challenge in the field, warranting further exploration.

## 5.3 SEARCH FOR THE OPTIMAL COEFFICIENT PARAMETER

We conduct an extensive exploration to determine the optimal coefficient parameter, denoted as $\lambda$, trying to strike a balance between adversarial robustness and classification accuracy on clean images. The investigation specifically targets the CIFAR-10 dataset and employs the WRN-16 architecture.

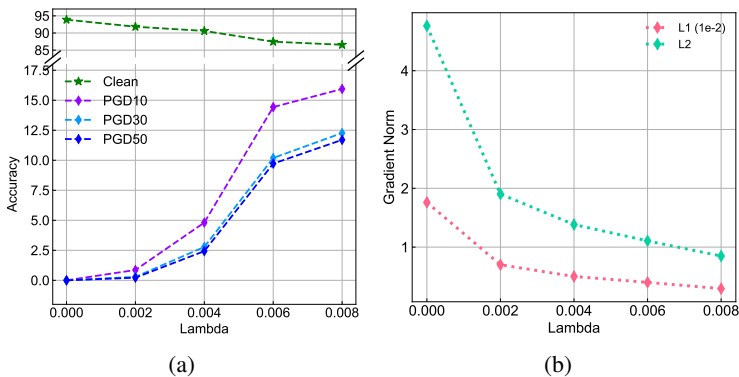

(a)                              (b)

Figure 2: The influence of the coefficient parameter $\lambda$ on classification accuracy and gradient sparsity. (a): Fluctuations in clean accuracy and adversarial accuracy under PGD attacks across different values of $\lambda$. (b): The $\ell_1$ and $\ell_2$ norms of the gradient with varying $\lambda$.

As described in Figure 2(a), we test the impact of varying the coefficient parameter within the range of 0.000 to 0.008. Notably, increasing the value of $\lambda$ led to a decrease in clean accuracy but a

significant improvement in adversarial robustness. To be specific, when using a coefficient parameter of $\lambda = 0.008$, the PGD10 adversarial accuracy increased from zero to 16%, while maintaining over 85% accuracy on clean images.

Figure 2(b) provides a visual representation of the effect of $\lambda$ on gradient sparsity after training. We computed the average $\ell_1$ and $\ell_2$ norm of $\nabla_{\boldsymbol{x}} f_y$ over the test dataset using the pre-trained model. Both the $\ell_1$ and $\ell_2$ norm values decrease significantly as the coefficient parameter increases, indicating the effectiveness of gradient sparsity regularization.

Based on these findings, we select $\lambda = 0.008$ to train the vanilla SR-WRN-16 on CIFAR-10 to strike a balance between clean accuracy and adversarial robustness. It is worth noting that the optimal choice of $\lambda$ may vary for different datasets. For additional insights into the relationship between $\lambda$, clean accuracy, and adversarial accuracy on CIFAR-100, please refer to the line chart presented in Appendix E.

### 5.4 VISUALIZATION OF GRADIENT SPARSITY

To validate the effectiveness of the proposed approximation method (Proposition 1), we compute the values of $\nabla_{\boldsymbol{x}} f_y(\boldsymbol{x})$ for $\boldsymbol{x}$ in three cases: $f$ is a vanilla SNN, $f$ is an adversarial trained SNN, and when $f$ is an SNN trained with the proposed gradient sparsity regularization. Figure 3 illustrates the distribution of $\nabla_{\boldsymbol{x}} f_y(\boldsymbol{x})$ for $\boldsymbol{x}$ across both CIFAR-10 and CIFAR-100.

The results clearly show that the distribution of gradient value for SR*-SNNs is more concentrated around zero compared to that of vanilla SNNs and AT-SNNs. This indicates that SR*-SNNs exhibit sparser gradients concerning the input image, demonstrating the effectiveness of the finite difference (Proposition 1) in constraining gradient sparsity. Meanwhile, these findings suggest a correlation between the gradient sparsity and SNN robustness to some extent: sparser gradients contribute to the enhancement of SNN robustness. Additional visual evidence in the form of heatmaps of $\nabla_{\boldsymbol{x}} f_y(\boldsymbol{x})$ for selected examples in CIFAR-10 is provided in Appendix F, further demonstrating that the gradients of SR-SNNs with respect to input images are sparser compared to those of vanilla SNNs.

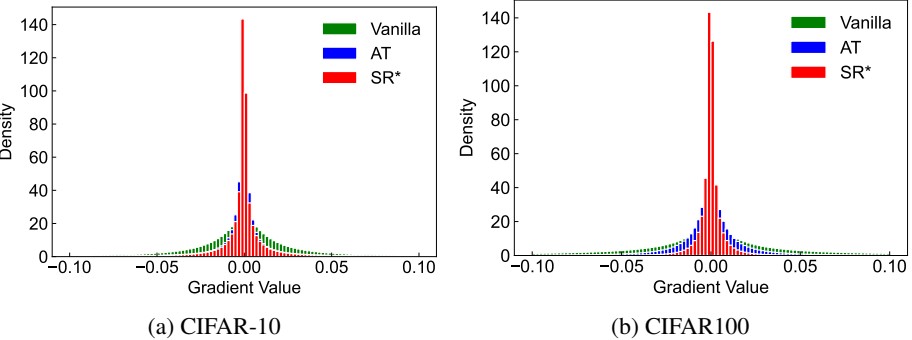

Figure 3: The normalized distribution of $\nabla_{\boldsymbol{x}} f_y(\boldsymbol{x})$.

## 6 CONCLUSION

This paper delves into a fresh perspective on SNN robustness by introducing the gradient sparsity. We theoretically proved that the ratio of adversarial vulnerability to random vulnerability of the SNN is upper bounded by the sparsity of the gradient of the output with respect to the input image. This insight sparked interest in exploring the robustness of event-driven SNNs.

**Limitations and future work.** The improvement of adversarial robustness by SR comes at the cost of a notable accuracy loss on clean images. Besides, the capability of the ensemble attack need to be further explored. In future work, we aim to strike a better balance between classification accuracy and adversarial robustness. For instance, we may explore employing the simulated annealing algorithm in SR* to dynamically adjust the weight of the regularization term. Moreover, we will further explore the impact of the width and shape of the surrogate function to the ensemble attack.

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

# A   PROOF OF THEOREM 1

**Theorem 1.** *Suppose $f$ represents an SNN and $\epsilon$ is the strength of an attack. Given an input image $\boldsymbol{x}$ with corresponding label $y$, the ratio of $\rho_{adv}(f, \boldsymbol{x}, \epsilon, \ell_{\infty})$ and $\rho_{rand}(f, \boldsymbol{x}, \epsilon, \ell_{\infty})$ is upper bounded by the sparsity of $\nabla_{\boldsymbol{x}} f_y$:*

$$3 \leqslant \frac{\rho_{adv}(f, \boldsymbol{x}, \epsilon, \ell_{\infty})}{\rho_{rand}(f, \boldsymbol{x}, \epsilon, \ell_{\infty})} \leqslant 3\|\nabla_x f_y(\boldsymbol{x})\|_0. \tag{13}$$

*Proof.* We assume $f$ to be differentiable, where the surrogate gradient is used. When $\epsilon$ is small, we can expand $f_y(\boldsymbol{x} + \epsilon \cdot \delta)$ at $f(\boldsymbol{x})$ by the first-order Taylor expansion

$$f_y(\boldsymbol{x} + \epsilon \cdot \delta) \approx f_y(\boldsymbol{x}) + \epsilon \nabla f_y(\boldsymbol{x})^T \delta. \tag{14}$$

So, $f_y(\boldsymbol{x} + \epsilon \cdot \delta) - f_y(\boldsymbol{x}) \approx \epsilon \nabla f(\boldsymbol{x})^T \delta$. As $\delta \in \mathbb{R}^m$ and $\delta_i \sim Unif([-1, 1])$, we have

$$\mathbb{E}(\delta_i \delta_j) = \begin{cases} 0 & i \neq j \\ \dfrac{1}{3} & i = j. \end{cases} \tag{15}$$

Therefore, $\rho_{\text{rand}}(f, \boldsymbol{x}, \epsilon, \ell_{\infty})$ can be approximated as follows:

$$\begin{aligned} \rho_{\text{rand}}(f, \boldsymbol{x}, \epsilon, \ell_{\infty}) &= \mathbb{E}_{\delta \sim Unif(cube)} \left(f_y(\boldsymbol{x} + \epsilon \cdot \delta) - f_y(\boldsymbol{x})\right)^2 \\ &\approx \epsilon^2 \nabla f_y(\boldsymbol{x})^T \mathbb{E}_\delta(\delta \delta^T) \nabla f_y(\boldsymbol{x}) \\ &= \frac{1}{3} \epsilon^2 \|\nabla f(\boldsymbol{x})\|_2^2. \end{aligned} \tag{16}$$

On the other hand,

$$\begin{aligned} \rho_{\text{adv}}(f, \boldsymbol{x}, \epsilon, \ell_{\infty}) &= \sup_{\delta \sim Unif(cube)} \left(f_y(x + \epsilon \cdot \delta) - f_y(\boldsymbol{x})\right)^2 \\ &\approx \epsilon^2 \left(\sup_{\delta \sim Unif(cube)} |\nabla f_y(\boldsymbol{x})^T \delta|\right)^2 \\ &= \epsilon^2 \left(\nabla f_y(\boldsymbol{x})^T \text{sign}(\nabla f_y(\boldsymbol{x}))\right)^2 \\ &= \epsilon^2 \|\nabla f_y(\boldsymbol{x})\|_1^2. \end{aligned} \tag{17}$$

Consequently, the gap between $\rho_{\text{adv}}(f, \boldsymbol{x}, \epsilon, \ell_{\infty})$ and $\rho_{\text{rand}}(f, \boldsymbol{x}, \epsilon, \ell_{\infty})$ can be measured by

$$\frac{\rho_{\text{adv}}(f, \boldsymbol{x}, \epsilon, \ell_{\infty})}{\rho_{\text{rand}}(f, \boldsymbol{x}, \epsilon, \ell_{\infty})} \approx 3 \frac{\|\nabla f_y(\boldsymbol{x})\|_1^2}{\|\nabla f_y(\boldsymbol{x})\|_2^2}, \tag{18}$$

which can be bounded by

$$3 \leq \frac{\rho_{\text{adv}}(f, \boldsymbol{x}, \epsilon, \ell_{\infty})}{\rho_{\text{rand}}(f, \boldsymbol{x}, \epsilon, \ell_{\infty})} \leqslant 3\|\nabla f_y(\boldsymbol{x})\|_0. \tag{19}$$

[proof of the inequality] For an $m$-dimensional vector $u \in \mathbb{R}^m$, we have $\|u\|_1 \geqslant \|u\|_2$. Because

$$\begin{aligned} \|u\|_1^2 &= \left(\sum_{i=1}^m |u_i|\right)^2 \\ &= \sum_{i=1}^m u_i^2 + \sum_i \sum_{j \neq i} |u_i u_j| \\ &\geqslant \sum_{i=1}^m u_i^2 = \|u\|_2^2. \end{aligned} \tag{20}$$

Moreover, let $a \in \mathbb{R}^m$ be an $m$-dimensional vector with $a_i = \text{sign}(u_i)$ Then

$$
\begin{aligned}
\|u\|_1 = \sum_{i=1}^{m} |u_i| &= \sum_{i=1}^{m} u_i a_i \\
&\leqslant \left( \sum_{i=1}^{m} u_i^2 \right)^{\frac{1}{2}} \left( \sum_{i=1}^{m} a_i^2 \right)^{\frac{1}{2}} \quad \text{(Cauchy Schwartz inequality)} \\
&= \|u\|_2 \sqrt{\|u\|_0}
\end{aligned}
\tag{21}
$$

$\square$

## B  PROOF OF PROPOSITION 1

**Proposition 1.** *Let $d$ be the signed input gradient direction: $d = sign(\nabla_{\boldsymbol{x}} f_y(\boldsymbol{x}, w))$. Let $h$ be the finite difference step size. Then, the $\ell_1$ gradient norm is approximated by*

$$
\|\nabla_{\boldsymbol{x}} f_y(\boldsymbol{x}, w)\|_1 \approx \left| \frac{f_y(\boldsymbol{x} + h \cdot d, w) - f_y(\boldsymbol{x}, w)}{h} \right|.
\tag{22}
$$

*Proof.* The first order Taylor estimation of $f_y(\boldsymbol{x} + h \cdot d, w)$ at point $\boldsymbol{x}$ is

$$
f_y(\boldsymbol{x} + h \cdot d, w) \approx f_y(\boldsymbol{x}, w) + h \cdot \nabla_{\boldsymbol{x}} f_y^T(\boldsymbol{x}, w) d.
\tag{23}
$$

Since $d = sign(\nabla_{\boldsymbol{x}} f_y(\boldsymbol{x}, w))$, we have

$$
\nabla_{\boldsymbol{x}} f_y^T(\boldsymbol{x}, w) d = \|\nabla_{\boldsymbol{x}} f_y(\boldsymbol{x}, w)\|_1.
\tag{24}
$$

Therefore,

$$
f_y(\boldsymbol{x} + h \cdot d, w) \approx f_y(\boldsymbol{x}, w) + h\|\nabla_{\boldsymbol{x}} f_y(\boldsymbol{x}, w)\|_1,
\tag{25}
$$

and $\|\nabla_{\boldsymbol{x}} f_y(\boldsymbol{x}, w)\|_1$ can be approximated by

$$
\left| \frac{f_y(\boldsymbol{x} + h \cdot d, w) - f_y(\boldsymbol{x}, w)}{h} \right|
\tag{26}
$$

$\square$

## C  TRAINING SETTINGS

We use the same training settings for all architectures and datasets. Our data augmentation techniques include RandomCrop, RandomHorizontalFlip, and zero-mean normalization. During training, we use the CrossEntropy loss function and Stochastic Gradient Descent optimizer with momentum. The learning rate $\eta$ is controlled by the cosine annealing strategy Loshchilov & Hutter (2017). We utilize the Backpropagation Through Time (BPTT) algorithm with a triangle-shaped surrogate function, as introduced by Esser et al. (2016). When incorporating sparsity gradient regularization, we set the step size of the finite difference method to 0.01. Also, we use a $\lambda = 0.002$ on CIFAR-10 and $\lambda = 0.001$ on CIFAR-100 for SR* method. For vanilla SR, we set $\lambda = 0.008$ on CIFAR-10 and $\lambda = 0.002$ on CIFAR-100. Adversarial training is implemented using the PGD method with an epsilon value of 2/255. The step size for the PGD method is set to 0.01, and the total step is 5. The detailed training hyper-parameters are listed below.

Table 3: Detailed training setting

| Initial LR | Batchsize | Weight Decay | Epochs | Momentum | $h$ | PGD-stepsize | PGD-step |
|---|---|---|---|---|---|---|---|
| 0.1 | 64 | 5e-4 | 200 | 0.9 | 0.01 | 0.01 | 5 |

## D    EVALUATION SETTINGS

As mentioned in the main text, we consider an ensemble attack approach for SNNs. This involves utilizing a diverse set of surrogate gradients and considering both STBP-based and RGA-based attacks. We conduct the following attacks as the ensemble attack. We consider an ensemble attack to be successful for a test sample as long as the model is fooled with any of the attacks from the ensemble.

- **STBP-based attack with triangle-shaped surrogate function**

$$\frac{\partial s_i^l(t)}{\partial u_i^l(t)} = \frac{1}{\gamma} \left|\left|\gamma - \left|u_i^l(t) - \theta\right|\right|\right|. \tag{27}$$

Here the hyper-parameter $\gamma$ is set to 1 (Esser et al., 2016).

- **STBP-based attack with sigmoid-shaped surrogate function**

$$\frac{\partial s_i^l(t)}{\partial u_i^l(t)} = \frac{1}{1 + \exp\left(-\gamma(u_i^l(t) - \theta)\right)} \tag{28}$$

Here the hyper-parameter $\gamma$ is set to 4.

- **STBP-based attack with arc tangent surrogate function.**

$$\frac{\partial s_i^l(t)}{\partial u_i^l(t)} = \frac{\gamma}{2(1 + (\frac{\pi}{2}\gamma(u_i^l(t) - \theta))^2)} \tag{29}$$

Here the hyper-parameter $\gamma$ is set to 2.

- **RGA-based attack with rate-based gradient estimation.** The hyper-parameter setting follows the paper (Bu et al., 2023).

## E    COEFFICIENT PARAMETER SEARCH ON CIFAR-100

Figure 4(a) shows the relationship between the coefficient parameter $\lambda$, clean accuracy, and adversarial accuracy on the CIFAR-100 dataset. To make a trade-off between classification accuracy on clean images and adversarial images, we choose $\lambda = 0.002$ for the vanilla SR-WRN-16 on CIFAR-100. Figure 4(b) illustrates that the $\ell_1$ and $\ell_2$ norm of $\nabla_{\boldsymbol{x}} f_y$ decrease significantly with the increase of $\lambda$.

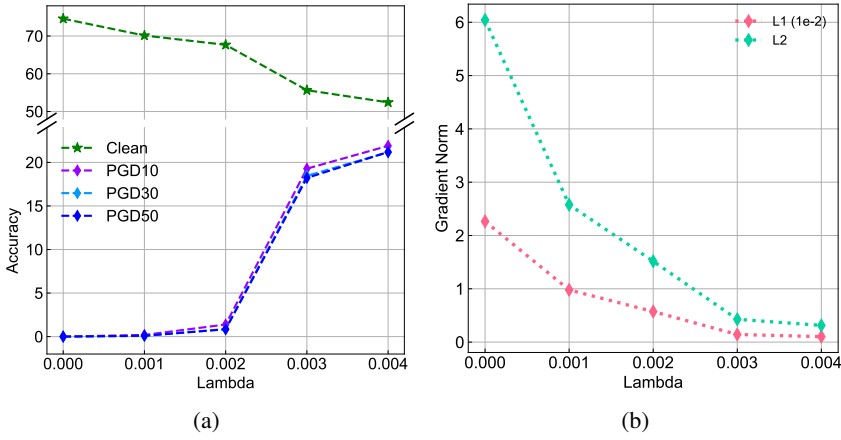

(a)                                                (b)

Figure 4: The influence of the coefficient parameter $\lambda$ on classification accuracy and gradient sparsity. (a): Fluctuations in clean accuracy and adversarial accuracy under PGD attacks across different values of $\lambda$. (b): The $\ell_1$ and $\ell_2$ norms of the gradient with varying $\lambda$.

## F    VISUALIZATION OF GRADIENT SPARSITY

To further substantiate the claim that SR-SNNs possess sparser gradients compared to vanilla SNNs, we present heatmaps of $\nabla_{\boldsymbol{x}} f_y$ for several examples in CIFAR-10 Tsipras et al. (2019). In Figure 5, the first row displays the original images from CIFAR-10, while the second and third rows show the corresponding heatmaps of $\nabla_{\boldsymbol{x}} f_y$ for vanilla SNN and SR-SNN, respectively.

Notably, the gradient of SR-SNN is sparser than that of the villain SNN. Moreover, the heatmap of gradients of the vanilla SNN appears cluttered to reflect any information about the image. However, the heatmap of the gradient of the SR-SNN shows some clear texture information of the image, which is beneficial to the interpretability of SNN. Therefore, we infer that the gradient sparsity regularization can not only improve the robustness of SNNs, but also provide some interpretability for SNNs.

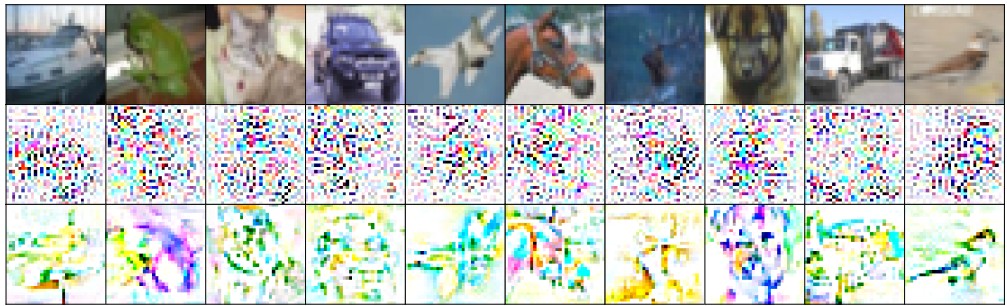

Figure 5: Visualization of $\nabla_{\boldsymbol{x}} f_y$ of a villain SNN and SR-SNN.

## G    DIFFERENCES BETWEEN SR AND ONE-STEP FGSM ADVERSARIAL TRAINING

In Theorem 1, we have proved that the disparity between the adversarial vulnerability and random vulnerability is upper bounded by the sparsity of $\nabla_{\boldsymbol{x}} f_y$. The main idea of our SR strategy is to regularize the value of $\|\nabla_{\boldsymbol{x}} f_y(\boldsymbol{x}, w)\|_0$. **It is important to note that in our approach,** $f(\cdot, w)$ **represents the model function, which differs from previous methods that utilize the multi-class calibrated loss function as a regularization term.**

To achieve this, we use $\|\nabla_{\boldsymbol{x}} f_y(\boldsymbol{x}, w)\|_1$ to approximate the $\ell_0$ norm of the gradient and then use finite difference method to make the regularization of $\|\nabla_{\boldsymbol{x}} f_y(\boldsymbol{x}, w)\|_1$ computationally feasible. Theoretically, if direct backpropagation on $\|\nabla_{\boldsymbol{x}} f_y(\boldsymbol{x}, w)\|_1$ were feasible, finite differences could be bypassed. However, as reported in Finlay & Oberman's work Finlay & Oberman (2021), directly calculating the backpropagation of $\|\nabla_{\boldsymbol{x}} f_y(\boldsymbol{x}, w)\|_1$ on ANNs is very time-consuming. Additionally, we found that SNNs are not trainable with such double backpropagation approach.

As depicted in Algorithm 1 (Line 5 to Line 7), $\hat{x}^i = x^i + h d^i$ is fundamentally different from an adversarial example in terms of both numerical value and meanings. Actually, it is used to calculate the difference quotient

$$\left| \frac{(f_{y^i}(\hat{\boldsymbol{x}}^i, w)) - (f_{y^i}(\boldsymbol{x}^i, w))}{h} \right| \tag{30}$$

which is the approximation of $\|\nabla_{\boldsymbol{x}^i}(f_{y^i}(\boldsymbol{x}^i, w))\|_1$. While theoretically, $d^i$ can be arbitrary, it is set to be the sign of the gradient in this paper. This choice aligns with the objective of approximating the $\ell_1$ norm of the gradient, as explained in detail in Appendix B.

Additionally, $h$ represents the step size used for the approximation, rather than the attack strength. It is recommended to choose a small value for $h$ in Eq. (30) to achieve a more accurate estimation of the $\ell_1$ norm. As $h$ increases, the approximation results become less accurate, leading to a decrease in robustness. However, if $h$ were the attack strength in one-step FGSM, larger $h$ would contribute to better robustness (at the expense of decreased accuracy on clean images). The impact of $h$ in our training algorithm is different from the impact of $h$ as the attack strength in FGSM adversarial

training. **In summary, the SR algorithm does not involve the calculation of adversarial samples, making it fundamentally different from adversarial training and its variations.**

# H DIFFERENCES WITH RELATE WORKS

## H.1 DIFFERENCES BETWEEN THE SR STRATEGY AND THE GRADIENT REGULARIZATION STRATEGY

The differences of the regularization term in this paper and the Gradient Regularization (GR) proposed by Finlay & Oberman (2021) lies in two key points.

The primary distinction lies in the fact that in SR, the regularization term is exclusively tied to the **model itself**, as opposed to being connected to the **multi-class calibrated loss function** used during the training phase. To provide further clarification, we can express the training loss of GR as follows (defending against $\ell_\infty$ attacks):

$$\min_w \left( l(x, w) + \frac{\lambda}{2} \|\nabla_x l(x, w)\|_1^2 \right). \tag{31}$$

where $l(x, w)$ is the multi-class calibrated loss function for the classification task (such as cross-entropy loss), $y$ is the label of the input $x$. In contrast, the training loss of our work is

$$\min_w \left( l(x, w) + \lambda \|\nabla_x f(x, w)\|_0 \right). \tag{32}$$

While Finlay & Oberman' work primarily focuses on the loss function, it is important to note that the regularization term in their approach varies depending on the specific choices of $l(x, w)$. In contrast, our proposed method introduces a regularization term that remains independent of the choice of loss function. Moreover, our method can be combined with adversarial training to enhance the performance of adversarial training (refers to Table 2 of the main text).

The objectives of these two training algorithms are different. In Finlay & Oberman's work, Eq. (31) acts as an upper bound for the loss function in adversarial training. Their goal is to minimize the loss of adversarial training through Eq. (31) and enhance the robustness of models. On the other hand, in our work, $\|\nabla_x f(x, w)\|_0$ represents an upper bound for the gap between the adversarial robustness and random robustness. We aim to reduce this gap and thereby improve the overall robustness of SNNs. Our objective is not directly associated with adversarial training.

## H.2 DIFFERENCES BETWEEN THE SR STRATEGY AND THE CLEAN LOGIT PAIRING

The clean logit pairing approach was proposed by Kannan et al. (2018). The differences between the proposed SR strategy and clean logit pairing lie in two key points.

Firstly, the regularization terms are different. The clean logit pairing method utilizes a regularization term $\|f(\boldsymbol{x}, w) - f(\boldsymbol{x}', w)\|_2^2$, where $\boldsymbol{x}$ and $\boldsymbol{x}'$ are two randomly selected clean images. In contrast, the regularization term in our paper, $\|\nabla_{\boldsymbol{x}} f_y(\boldsymbol{x}, w)\|_0$, is theoretically related only to the input image $\boldsymbol{x}$. It is important to note that the approximation formula $|\frac{f_y(\boldsymbol{x}, w) - f_y(\boldsymbol{x} + hd, w)}{h}|$ involves two closely situated points, namely $\boldsymbol{x}$ and $\boldsymbol{x} + hd$, rather than two randomly chosen images.

Secondly, the purposes of the regularization term are different. The clean logit pairing approach aims to promote similarity in probability distributions for two randomly chosen clean images. This method improves models' robustness on MNIST and SVHN, although it lacks theoretical guarantees. In contrast, the SR strategy in our paper aims to improve the sparsity of the input gradient $\nabla_{\boldsymbol{x}} f_y(\boldsymbol{x}, w)$ to reduce the gap between adversarial robustness and random robustness. The SR strategy is supported by theoretical analysis (Theorem 1).

## H.3 DIFFERENCES BETWEEN THE SR STRATEGY AND THE ADVERSARIAL LOGIT PAIRING/ TRADES

Adversarial logit training was proposed by Kannan et al. (2018), and TRADES is an extension version of it Zhang et al. (2019). Both adversarial logit pairing approach (RFGSM-40) and TRADES (solving a maximum problem) utilize adversarial examples during their training phase. However, it

is important to clarify that the proposed SR strategy does not use adversarial examples as explained in Appendix G. In other words, SR is not a kind of adversarial training.

The objectives are different. The primary objective of the adversarial logit pairing and TRADES is to encourage the probability distributions of the input image and its corresponding adversarial example to be similar. In contrast, our proposed SR strategy aims to induce sparsity in the gradient $\nabla_{\boldsymbol{x}} f_y(\boldsymbol{x}, w)$. We are not targeting the regularization of the probability distribution, and the regularization term corresponds to the gradient of $f_y(\boldsymbol{x}, w)$, rather than the output vector $f(\boldsymbol{x}, w)$. From an intuitive perspective, minimizing the $\ell_0$ norm of $\nabla_{\boldsymbol{x}} f_y(\boldsymbol{x})$ serves to bring $\boldsymbol{x}$ closer to a local maximum point, making it harder for attackers to generate adversarial examples through gradient-based methods.

## I    COMPARISON IN COMPUTATIONAL COST

The computational costs for one epoch training of various algorithms, including vanilla, PGD5 AT, RAT, SR, and SR*(PGD5+SR), on the CIFAR-10 dataset using the VGG11 architecture are summarized in Table 4. From the table, we observe that single SR incurs a computational cost 1.5 times that of RAT but consumes less than half the time needed by PGD5 AT. The computational cost of SR* is the highest among all training algorithms since it combines both SR and AT. However, models trained with SR* acheive the best robustness compared to models trained with other methods.

Table 4: The computational cost in one epoch of different training algorithms

| Vanilla | PGD5 AT | RAT | SR | SR* |
|---------|---------|------|------|------|
| 65s | 459s | 134s | 193s | 583s |

## J    DERIVATION OF EQUATION (10)

Let $\boldsymbol{x}$ denote the image, and $\{\boldsymbol{x}[1], \boldsymbol{x}[2], \ldots, \boldsymbol{x}[T]\}$ represent the input image series. In our paper, we use $\boldsymbol{x}[t] = \boldsymbol{x}$ for all $t = 1, \ldots, T$. The network is denoted by $f$ with parameters $w$. The output of the network $f$ is a vector $f(\boldsymbol{x}, w) \in \mathbb{R}^{N \times 1}$, where $N$ represents the number of classes. The output vector $f(\boldsymbol{x}, w)$ is obtained by applying the softmax function to the output of the last layer, i.e.

$$f(\boldsymbol{x}, w) = \text{softmax}\left(\sum_{t=1}^{T} s_1^L[t], \ldots, \sum_{t=1}^{T} s_N^L[t]\right). \tag{33}$$

Therefore, the $y^{th}$ component of $f(x, w)$ where $y$ represents the true label is

$$f_y(\boldsymbol{x}, w) = \text{softmax}_y\left(\sum_{t=1}^{T} s_1^L[t], \ldots, \sum_{t=1}^{T} s_N^L[t]\right). \tag{34}$$

According to the chain rule, the gradient of $f_y$ with respect to the input $\boldsymbol{x}$ is

$$\nabla_{\boldsymbol{x}} f_y(\boldsymbol{x}, w) = \frac{\partial \text{softmax}_y}{\partial \sum_t s_1^L[t]} \cdot \nabla_{\boldsymbol{x}}\left(\sum_t s_1^L[t]\right) + \cdots + \frac{\partial \text{softmax}_y}{\partial \sum_t s_N^L[t]} \cdot \nabla_{\boldsymbol{x}}\left(\sum_t s_N^L[t]\right). \tag{35}$$

In this formula, the gradient of the output the $i^{th}$ component $\nabla_{\boldsymbol{x}}\left(\sum_t s_i^L[t]\right)$ in the last layer with respect to $x$ can be further expresses as

$$\nabla_{\boldsymbol{x}}\left(\sum_t s_i^L[t]\right) = \sum_t \nabla_{\boldsymbol{x}} s_i^L[t] = \sum_t \sum_{\tilde{t}=1}^{t} \nabla_{\boldsymbol{x}[\tilde{t}]} s_i^L[t]. \tag{36}$$

Finally, the gradient $\nabla_{\boldsymbol{x}} f_y(x, w)$ is written as

$$\nabla_{\boldsymbol{x}} f_y(\boldsymbol{x}, w) = \sum_{i=1}^{N}\left(\frac{\partial \text{softmax}_y}{\partial \sum_t s_i^L[t]}\left(\sum_{t=1}^{T} \sum_{\tilde{t}=1}^{t} \nabla_{\boldsymbol{x}[\tilde{t}]} s_i^L[t]\right)\right). \tag{37}$$