# OpenReview forum: "Training Adversarially Robust SNNs with Gradient Sparsity Regularization"
_ICLR.cc/2024/Conference — Submitted to ICLR 2024_

### Official Review · Reviewer_FfwP · 2023-10-24

**Soundness:** 2 fair
**Presentation:** 2 fair
**Contribution:** 2 fair
**Rating:** 5
**Confidence:** 4

**Summary:**

This paper investigates the robustness of SNNs against adversarial perturbations. An initial robustness analysis reveals that SNNs are robust against random perturbations, but vulnerable against adversarial attacks. The proposed method incorporates the gradient sparsity regularization in the loss function to reduce the gap between the SNN robustness against random noise and adversarial perturbations. The experiments of the proposed method conducted on CIFAR-10 and CIFAR-100 dataset reveal higher SNN robustness compared to the traditional approach.

**Strengths:**

1. The tackled problem is relevant to the community.

2. The proposed method is original.

3. The experimental results show higher robustness of the proposed method compared to prior works.

**Weaknesses:**

There are some aspects to clarify and improve. Please see the questions below.

**Questions:**

1. Please discuss more in detail what are the key findings made in the existing literature in terms of SNN robustness, what are the limitations of the existing methods, and how the challenges are solved in this paper.

2. Referring to Figure 1, what are the experiment settings used to generate the results? What is the SNN architecture? What is the adversarial attack algorithm?

3. In Section 4.3, please discuss more clearly the differences between the approximation of the gradient regularization term employed in this paper and the related works.

4. In Section 5, please discuss in detail all the parameters and hyperparameters used to conduct the experiments, as well as the tool flow. If possible, please provide the code in an online open-source repository.

5. From Table 1 we can infer that, while the proposed method can improve the adversarial robustness, there is a significant accuracy loss for clean inputs compared to related works. Please discuss the limitations and potential solutions to overcome this issue.

6. The experiments have been conducted only on CIFAR-10 and CIFAR-100 dataset. It is recommended to make experiments also on event-based datasets, which are typical benchmarks for SNNs.

---

> ### Author Response · Authors · 2023-11-16
> **Response to Reviewer FfwP (1/2)**
>
> Thanks for the valuable comments. We are encouraged that you find the
> tackled problem is relevant to the community, our work original and our experimental results good. We would like to address your concerns and your questions in the following.
>
> **Q1: Discuss more in detail the key findings and limitations in the existing literature, and how the challenges are solved in this paper.**
>
> Thanks for your valuable feedback! The key findings and limitations in the existing literature are as follows. Methods for improving the robustness of SNNs can be broadly categorized into two classes. The first class draws inspiration from ANNs. A typical representative is adversarial training and its variants. This approach has been shown to effectively defend against attacks that are used in the training phase. Another method is certified training, whose application to SNNs remains challenging. Current efforts in this area have primarily focused on the MNIST dataset. The second category consists of SNN-specific techniques designed to enhance robustness, such as using the Poisson encoder. However, the Poisson encoder generally yields worse accuracy on clean images than the direct encoding, and the robustness improvement caused by the Poisson encoder varies with the number of time-steps used.
>
> In this paper, we propose the SR strategy which can be combined with adversarial training to boost the robustness of SNNs. The SR strategy is supported by a solid theoretical foundation, and experimental results on CIFAR-10 and CIFAR-100 demostrate its effectiveness in enhancing the robustness of SNNs.
>
> We have revised Sec. 2.2 to provide a more detailed discussion of the key findings and limitations in the existing literature in terms of SNN robustness, as well as how the challenges are solved in this paper.
>
> **Q2. The experiment settings used to generate Figure 1.**
>
> The experimental settings in Figure 1 are outlined as follows. The SNN architecture is a VGG-11 model. The adversarial attack employed is FGSM, with $\epsilon$ ranging from $0.00$ to $0.25$. For the random attack on SNNs, a perturbation $\delta$ is randomly generated to satisfy a uniform distribution within the hypercube $\Vert \delta \Vert_\infty \leqslant \epsilon$. Subsequently, $\delta$ is added to the input image $x$ and the pixel value of the perturbed image are clipped to the range of [0, 1]. The classification accuracy is evaluated for all $x+\delta$, where $x$ is drawn from the test set of CIFAR-10 (Figure 1(a)) and CIFAR-100 (Figure 1(b)). We have added these detailed settings in Sec. 4.1 of the revised version.
>
> **Q3: Discuss more clearly the differences between the approximation of the gradient regularization term employed in this paper and the related works.**
>
> Regarding the approximation method for computing the gradient norm, both our paper and the related work ([Finlay & Oberman 2021](https://www.sciencedirect.com/science/article/pii/S2666827020300177)) use the finite differences technique. The only difference is that, we take the sign of the gradient as the difference direction while Finlay & Oberman took the sign direction divided by $\sqrt{N}$, where $N$ is the dimension of $x$. For other differences between the objectives of regularization terms in this paper and the related work, please refer to "To All Reviewers".
>
> **Q4: Discuss in detail all hyperparameters used to conduct the experiments, and provide the code.**
>
> We would like to clarify that the setting of hyperparameters are discussed in Appendix C and D. Thanks for your suggestion! We have already uploaded the code in the revised supplementary material.
>
> **Q5: Please Discuss the limitations and potential solutions to overcome accuracy loss for clean inputs casued by SR.**
>
> Thanks for the valuable suggestion! The enhancement of adversarial robustness through SR is accompanied by a notable loss in accurac on clean images. In future work, we aim to achieve a better trade-off between classification accuracy and adversarial robustness. One possible avenue we plan to explore is the utilization of the simulated annealing algorithm in SR* to dynamically adjust the weight assigned to the regularization term, potentially leading to a more balanced outcome in terms of both accuracy and robustness.
>
> We have added these limitations and potential solutions in Section 6 of the revised paper.

---

> > ### Author Response · Authors · 2023-11-21
> > **Response to Reviewer FfwP (2/2)**
> >
> > **Q6: Experiments on DVS.**
> >
> > We use the VGG-11 on DVS-CIFAR10 Dataset and compared Vanilla trained model and SR trained model. When evaluating, we perform the adversarial attack on preprocessed frames and add adversarial noise onto each frame. The robustness performance is shown in Table R8. The performance of SR trained model is significantly higher than the Vanilla model, which proves the effectiveness of SR method.
> >
> > **Table R8. Performance of models on DVS-CIFAR10**
> > | Methods    | Clean | FGSM  | PGD10 | PGD30 | PGD50 |
> > |------------|-------|-------|-------|-------|-------|
> > | Vanilla    | 71.50 | 24.50 | 10.10 |  9.50 |  9.40 |
> > | SR         | 71.70 | 42.00 | 34.10 | 34.30 | 33.60 |

---

> > > ### Comment · Reviewer_FfwP · 2023-11-23
> > > **Response to Authors' Rebuttal**
> > >
> > > Thank you for your responses. Considering together the other reviews and responses, my score is confirmed.

---

> ### Author Response · Authors · 2023-11-21
> **Kindly Reminder**
>
> Dear reviewer FfwP, thanks again for your careful and valuable comments! Since the rebuttal discussion is due soon, we’ll be appreciated to know whether our replies have addressed your questions. If there are any further clarifications required or any other concerns, please feel free to contact us. Many thanks!

---

### Official Review · Reviewer_HCoc · 2023-10-24

**Soundness:** 1 poor
**Presentation:** 2 fair
**Contribution:** 1 poor
**Rating:** 1
**Confidence:** 5

**Summary:**

This paper focuses on the problem of improving adversarial robustness of SNNs. Authors propose an input gradient sparsity promoting regularization scheme for training robust SNNs. An l0-norm penalty term on the input gradient is approximated via a sparsity-promoting l1-norm penalty, which is then again approximated with a softmax output regularization term using the finite differences method. Proposed approach is then combined with a traditional adversarial training method for SNNs, and empirically showcased on CIFAR-10/100.

**Strengths:**

- Theoretical justification on why this method could be applicable for robustness is presented nicely.
- Writing is clear and the storyline is presented well.

**Weaknesses:**

- Robustness evaluations of the SNNs are ambiguous in a weak way, and need much more rigor & depth.
- Limited innovation and justification from the ML security methodology aspect, as well as the practical side.

**Questions:**

1) Can the authors clearly state/elaborate in their paper, in which ways their training algorithm is different than the work by [Finlay & Oberman 2019]? https://arxiv.org/pdf/1905.11468.pdf

2) Proposed approach has several simplifications/assumptions on the ultimate gradient l0-norm regularization idea. At the end, the used training algorithm seems to become also similar to the well-known (clean) logit pairing approach [Kannan, Kurakin & Goodfellow, 2018]. At the very least, the objective simply uses adversarial examples for an output probability distribution regularizer, which has been the most fundamental form of adversarial training to date (see e.g., [Zhang et al. ICML 2019]). Hence all in all, I can not clearly see any innovation from the ML security side in this paper. Can the authors experimentally compare why their choice would be particularly any better in the case of SNNs, than using one of the other powerful adversarial training/regularization methods (e.g., TRADES [Zhang et al. ICML 2019] or adversarial logit pairing [Kannan, Kurakin & Goodfellow, 2018])?

3) Authors claim to use an "ensemble attack" for white-box evaluations, which requires further clarification. Their implementation of an ensemble "conducts multiple attacks on each sample and reports the strongest attack". Can the authors provide an exemplary outlined test set case more clearly on how these evaluations are reported?

4) For this attack ensemble to be meaningful and reveal any impact of gradient obfuscation, there should actually be cases where the surrogate gradient function is changing its width, rather than only its shape for fixed parameters. The width parameter $\gamma=1$ of the triangular surrogate should be accounted for. Authors should run an extensive evaluation with for instance $\gamma\in[0.1,3.0]$ in fine-grained steps of 0.1, and demonstrate that changing this parameter does not at all influence the capability of the adversary any better than using different surrogate gradient shapes. This part overall needs strong empirical justification.

5) Models were trained using PGD-5 adversarial examples, but the Algorithm 1 denotes adv. examples obtained via one-step FGSM. Which one is correct? Does this mean that PGD-5 is the general AT approach adopted in SR* models, but in any case the regularizer term was always obtained via FGSM? Needs clarification overall.

6) Following my question above, Table 2 is a bit complicated in how SR and AT can be disentangled. In the case of ablation models without AT but SR, how were the adv. examples to compute the regularization term, obtained? None of these details are clear in the paper.

7) Proof of Thm 1 in Appendix A seems agnostic to any function f, regardless of being an SNN. However, it appears to implicitly make the assumption that the SNN uses direct input coding. Can the authors comment if these assumptions would also hold for SNNs that use Poisson input coding, or SNNs that do not necessarily use IF neurons (i.e.., with membrane potential leak)? Wouldn't then the error that should be accounted for in the finite differences approximation would be large? Did the authors experiment with other types of more realistic SNNs at all?

8) Did the authors perform any simulations on dynamic vision sensor data where SNNs are designed to be more beneficial for?

9) Since PGD-5 is used via BPTT during training for 200 epochs, than the authors should outline a table with the computational overhead (wall-clock time) of their approach, in comparison to simple AT or RAT with PGD-5 as well.

10) Figure 3 should also include adversarially trained models in comparison (not only Vanilla), since such SNNs are already implicitly inducing a similar behavior.

Minor comment: Eq (3) & (4) are already defined as solutions of an adversarial example in the l_infty norm (since the sign function is used). Therefore at the end of Sec 3.2, authors should correct the "l_p norm", since p is already infty in this setting.

---

> ### Author Response · Authors · 2023-11-16
> **Response to Reviewer HCoc (1/4)**
>
> Thank you for your valuable insights. We sincerely appreciate your advice regarding our work. We apologize for any ambiguities in our description of the SR method that may have caused misunderstandings. In this rebuttal, we would like to first clarify the distinction between SR and adversarial training. Subsequently, we will address your concerns and provide responses to your questions. If you have any further suggestions or questions, please feel free to let us know.
>
> **Q1: Clarification of the SR strategy**
>
> In Theorem 1 of the main text, we have proved that the disparity between the adversarial vulnerability and random vulnerability is upper bounded by the sparsity of $\nabla_x f_y$. The main idea of our SR strategy is to regularize the value of $\Vert \nabla_x f_y(x,w) \Vert_0$. **It is important to note that in our approach, $f(\cdot, w)$ represents the model function, which differs from previous methods that utilize the multi-class calibrated loss function as a regularization term.** Here, the multi-class calibrated loss function $l(f(x,w),y)$, such as cross-entropy, is the loss function used to train a classifier where $y$ is the true label of $x$.  In other words, our proposed method introduces a regularization term that remains independent of the choice of loss function.
>
> To achieve the regularization of $\Vert \nabla_x f_y(x,w) \Vert_0$, we use $\Vert \nabla_x f_y(x,w) \Vert_1$ to approximate the $\ell_0$ norm of the gradient and then use finite difference method to make the regularization of $\Vert \nabla_x f_y(x,w) \Vert_1$ computationally feasible. Theoretically, if direct backpropagation on $\Vert \nabla_x f_y(x,w) \Vert_1$ were feasible, finite differences could be bypassed. However, as reported in [Finlay & Oberman's work](https://arxiv.org/pdf/1905.11468.pdf), directly calculating the backpropagation of $\Vert \nabla_x f_y(x,w) \Vert_1$ on ANNs is very time-consuming. Additionally, we found that SNNs are not trainable with such double backpropagation approach.
>
>
> We also find that the demonstration of Algorithm 1 in the main text is not clear enough, potentially leading readers to confuse our proposed SR algorithm with the adversarial training method and its variants. We would like to clarify that, as depicted in Algorithm 1 (Line 5 to Line 7), $\hat{x}^i=x^i + hd^i$ is fundamentally different from an adversarial example in terms of both numerical value and meanings. Actually, it is used to calculate the difference quotient
> \begin{equation}
>  \left| \frac{
>         (f_{y^i}(\hat{x}^i, w)) - (f_{y^i}(x^i, w))
>         }{
>         h
>         } \right|
> \tag{3-1}
> \end{equation}
> which is the approximation of $\Vert \nabla_{x^i} (f_{y^i}(x^i, w)) \Vert_1$. While theoretically, $d^i$ can be arbitrary, it is set to be the sign of the gradient in this paper. This choice aligns with the objective of approximating the $\ell_1$ norm of the gradient, as explained in detail in Appendix B.
>
> Additionally, $h$ represents the step size used for the approximation, rather than the attack strength. It is recommended to choose a small value for $h$ in Equation (3-1) to achieve a more accurate estimation of the $\ell_1$ norm. As $h$ increases, the approximation results become less accurate, leading to a decrease in robustness. However, if $h$ were the attack strength in one-step FGSM, larger $h$ would contribute to better robustness (at the expense of decreased accuracy on clean images). The impact of $h$ in our training algorithm is different from the impact of $h$ as the attack strength in FGSM adversarial training. **In summary, the SR algorithm does not involve the calculation of adversarial samples, making it fundamentally different from adversarial training and its variations.**
>
> We have revised the parts of the manuscript that may cause misunderstandings and added this clarification in Appendix G.
>
> **Q2: Differences between our training algorithm and the work by [Finlay & Oberman 2019](https://arxiv.org/pdf/1905.11468.pdf).**
>
> Please refer to "To All Reviewers".
>
> **Q3: Differences between SR and (clean) logit pairing approach. Differences between SR and (adversarial) logit pairing approach, as well as TRADES.**
>
> Please refer to "To All Reviewers".

---

> ### Author Response · Authors · 2023-11-16
> **Response to Reviewer HCoc (2/4)**
>
> **Q4: Further clarification of an "ensemble attack" for white-box evaluations.**
>
> We would like to clarify that as dicussed in the Appendix D, to avoid gradient obfuscation, the ensemble attack involves utilizing a diverse set of attacks with different surrogate gradients and gradient backward methods. For each test image, we applied various attacks with the following setting. **We consider an ensemble attack to be successful for a test sample as long as the model is fooled by any of the attacks from the ensemble.**
>
> **Table R3. The experimental settings for the ensemble attack**
> | Attacks            | Attack 1 | Attack 2 | Attack 3   | Attack 4 |
> |--------------------|----------|----------|------------|----------|
> | Surrogate function | Triangle | Sigmoid  | Arctangent | Triangle |
> | hyperparameter     | 1        | 4        | 2          | /        |
> | Backward method    | STBP     | STBP     | STBP       | RGA      |
>
> **Q5: Extensive evaluation with for instance $\gamma \in [0.1,3.0]$.**
>
> Thanks for your valuable suggestion, we have applied the attack with $\gamma \in [0.1, 3.0]$ and Table R4 reported the results of a PGD10 attack on VGG-11 models with different training algorithms on the CIFAR-10 dataset.
>
> We compare three different attack combinations to evaluate the impact of different surrogate functions on the attack strength. We selecte RAT, PGD5-AT and SR*(PGD5+SR) model as the target models. For PGD10(w/o ensemble), we only use the Triangle-shaped surrogate function， which is identical to the one used in training. For PGD10 (w/ ensemble), we use the attack combination as described in the manuscript. For PGD10($\gamma$ from [0.1, 3.0]), we incorporate 30 different Triangle-shaped surrogate function with its $\gamma$ ranging from [0.1, 30], as suggested.
>
> We find that both ensemble attack methods significantly improve attack performance and mitigate the impact of gradient obfuscation. This indicates that both shape and width of the surrogate function can influence the capability of the adversary. In addition, we acknowledge that the PGD10($\gamma$ from [0.1, 3.0]) attack is slightly more effective than the ensemble attack used in this paper. However, it is important to note that the PGD10($\gamma$ from [0.1, 3.0]) attack uses a 30-fold fine-grained grid search over attack hyperparameters for each image, which is considerably more computationally expensive compared to the ensemble attack used in our paper.
>
> In conclusion, on one hand, we agree with reviewer HCoc's claim that both width and shape of the surrogate function can impact attack performance. We have included additional discussions about this in the limitation part. On the other hand, we would like to clarify that our contribution mainly lines in the defense algorithm, not the attack algorithm. In our paper, we use the same attack method for all comparisons to ensure a fair comparison of different methods. We believe that whether we use our current method or incorporate the attack combinations suggested by reviewer HCoc, it will not significantly influence the effectiveness of our defense strategy.
>
> **Table R4. The classification accuracy (%) under the ensemble attack with different &gamma;**
> | Attacks                   | RAT   | PGD-AT | SR*   |
> |---------------------------|-------|--------|-------|
> | PGD10 (w/o ensemble)      | 16.16 | 21.32  | 33.67 |
> | PGD10 (w/ ensemble)       | 11.53 | 18.18  | 30.54 |
> | PGD10 (γ from [0.1, 3.0]) | 11.87 | 16.16  | 27.06 |
>
> **Q6: Models were trained using PGD-5 adversarial examples, but the Algorithm 1 denotes adv. examples obtained via one-step FGSM. Which one is correct?**
>
> Thanks for pointing it out! We would like to clarify that in the SR model, we do not use adversarial examples obtained via one-step FGSM to conduct adversarial training, as explained in "Clarification of the SR strategy".  For SR* models, we employ PGD-5 as the general adversarial training approach, and the training loss function is
> \begin{equation}
>     \mathcal{L}(x^i, y^i,w) = \text{CE}(f(x_{adv}^i), y^i) +
>         \lambda_{\text{SR}}\left\Vert \nabla_{x_{adv}^i} f(x_{adv}^i,w)\right \Vert_1
>     \tag{3-2}
> \end{equation}
> where $\left\Vert \nabla_{x^i} f(x^i,w)\right \Vert_1$ is approximated by the finite differences, and $x_{adv}^i$ is obtained by PGD-5.

---

> ### Author Response · Authors · 2023-11-16
> **Response to Reviewer HCoc (3/4)**
>
> **Q7: Following my question above, Table 2 is a bit complicated in how SR and AT can be disentangled. In the case of ablation models without AT but SR, how were the adv. examples to compute the regularization term obtained?**
>
> As clarified in the "Clarification of the SR strategy" section, the finite difference approximation employed in SR is not a type of adversarial training using one-step FGSM. Thus, SR is not entangled with AT, and the ablation models without AT but with SR do not involve adversarial examples. To provide further clarity on the distinction between SR and AT, we present the training loss function used in different training algorithms.
>
> For PGD5-AT: \begin{equation}
>     \mathcal{L}(x^i, y^i,w) = \text{CE}(f(x_{adv}^i), y^i)
>     \tag{3-3}
> \end{equation}
> For single SR: \begin{equation}
>     \mathcal{L}(x^i, y^i,w) = \text{CE}(f(x^i), y^i) +
>         \lambda_{\text{SR}}\left\Vert \nabla_{x^i} f(x^i,w)\right \Vert_1
>         \tag{3-4}
> \end{equation}
> For SR*(SR+PGD5): \begin{equation}
>     \mathcal{L}(x^i, y^i,w) = \text{CE}(f(x_{adv}^i), y^i) +
>         \lambda_{\text{SR}}\left\Vert \nabla_{x_{adv}^i} f(x_{adv}^i,w)\right \Vert_1
>         \tag{3-5}
> \end{equation}
>
> **Q8: If assumptions in this paper would also hold for SNNs that use Poisson input coding, or SNNs that use LIF?**
>
> Thanks for pointing it out. Indeed, the SR strategy can be applied theoretically as long as SNN is differentiable through methods such as surrogate gradients.
>
> To demonstrate this point, we apply SR to SNNs with VGG-11 structure consisted of **LIF neurons** on the CIFAR-10 dataset. The experimental results shown in Table R5 clearly demonstrate the effectiveness of the SR* strategy in enhancing the robustness of SNNs with LIF neurons. Models trained with SR* exhibit significantly improved robustness against all PGD attacks compared to models trained with AT, with a 4% decline in classification accuracy on clean images.
>
> **Table R5-1. Experiments on SNNs with LIF neurons**
> | Models    | Clean | PGD10 | PGD30 | PGD50 |
> |-----------|-------|-------|-------|-------|
> | VGG11-AT  | 89.77 | 17.07 | 14.55 | 14.27 |
> | VGG11-SR* | 85.60 | 30.80 | 28.52 | 28.38 |
>
> We also trained two VGG-11 models on CIFAR-10 using **Poisson encoding**. During training, we use the same setting as reported in the main text with the only different being the input encoding. When evaluating robust performance, the Poisson input encoding can be treated as an input transformation and use the EOT method ([Athalye, Anish, Nicholas Carlini, & David Wagner](http://proceedings.mlr.press/v80/athalye18a.html)) to attack. According to the results in Table R5-2, the performance of SR model is higher than the Vanilla model and the robust performance of SNN trained with SR is even 15% higher than the vanilla SNN under PGD10 attack. That proves the effectiveness of the sparsity regularization.
>
> **Table R5-2. Experiments on SNNs using Poisson Encoding**
> | Methods    | Clean | FGSM  | PGD10 |
> |------------|-------|-------|-------|
> | Vanilla    | 82.71 | 26.58 | 17.69 |
> | SR         | 78.84 | 39.03 | 32.67 |
>
> These experiments demostrate that our strategy also holds for SNNs with LIF neurons or with Poisson encoding.
>
> **Q9: Experiments on DVS datasets**
>
> We use the VGG-11 architecture on DVS-CIFAR10 Dataset and compared Vanilla trained model and SR trained model. When evaluating, we perform the adversarial attack on preprocessed frames and add adversarial noise onto each frame. The robustness performance is shown in Table R6. The performance of SR trained model is significantly higher than the Vanilla model, which proves the effectiveness of SR method.
>
> **Table R6. Performance of models on DVS-CIFAR10**
> | Methods    | Clean | FGSM  | PGD10 | PGD30 | PGD50 |
> |------------|-------|-------|-------|-------|-------|
> | Vanilla    | 71.50 | 24.50 | 10.10 |  9.50 |  9.40 |
> | SR         | 71.70 | 42.00 | 34.10 | 34.30 | 33.60 |
>
> **Q10: The computational overhead of SR and SR\*, in comparison to simple AT or RAT with PGD-5 as well.**
>
> The computational costs for one epoch training of various algorithms, including vanilla, PGD5 AT, RAT, SR, and SR*(PGD5+SR), on CIFAR-10 using the VGG11 architecture are summarized in Table R6. From the table, we observe that single SR incurs a computational cost 1.5 times that of RAT but consumes less than half the time needed by PGD5 AT. The computational cost of SR* is the highest among all training algorithms since it combines both SR and AT. However, models trained with SR* acheive the best robustness compared to models trained with other methods. We have added this comparison in Appendix I.
>
> **Table R7. The computational cost in one epoch of different training algorithms**
> | Vanilla | PGD5 AT | RAT    |  SR    | SR*     |
> | ------- | ------- | -----  | ------ | ------- |
> | 65s     | 459s    | 134s   |  193s  | 583s    |

---

> ### Author Response · Authors · 2023-11-21
> **Thank you for the time and hope our responses helpful for your re-assessment of our work.**
>
> Dear reviewer HCoc, we sincerely hope our posted response can help to address your concerns on our paper and serve as a reference for your re-assessment of our work. If you have any further comments and questions, please let us know and we are glad to write a follow-up response.

---

> ### Author Response · Authors · 2023-11-21
> **Response to Reviewer HCoc (4/4)**
>
> **Q11: Include adversarially trained models in Figure 3.**
>
> We have added the gradient distribution of AT-trained models with respect to the input image in Figure 3. The distribution of gradient values for SNNs-SR* is more concentrated around zero compared to that of vanilla SNNs and SNNs-AT.
>
> **Q12: Minor comments**
>
> Thanks for pointing it out. We correct the "$\ell_p$ norm" at the end of Sec 3.2 in the revised version.

---

> > ### Comment · Reviewer_HCoc · 2023-11-22
> > **response to authors**
> >
> > Thanks to the authors for their clarifications and experiments.
> >
> > I think this work is not really convincing in its empirical contributions & implementations, as well as the particular novelty it introduces towards making SNNs more robust with reasonable trade-offs. Considering the other reviewers’ comments, scores and confidences, I will keep my initial recommendation to reject. Here are some further feedback for the authors’ from my side:
> >
> > - Given the results in Table R4, I believe there can be an adversary obfuscation present if one can simply manipulate the surrogate gradient parameter to attack better. Authors claim that the way attack combinations are implemented (as I did suggest) will not influence the effectiveness of their defense strategy. I disagree. Thorough evaluations are necessary in empirical defenses, and at any rate they might lead to more convincing results if defenses are reliable. More broadly, it influences the significance of the work for ML security. This work does not provide a certified robustness solution for SNNs. Since results are only empirical at CIFAR scale, the experiments and adversary evaluations should at least reveal worst-case performance (see [Carlini et al 2019], section “skepticism of results”).
> >
> > - Differences to adv. logit-based regularization methods: I now see the minor choice differences than existing defenses, which appears more clearly in formulas. But again, one should have than justified that this particular regularizer choice is specifically better for SNNs, than simply using TRADES with PGD-5 for instance. The novelty proposed is not really justified there.
> >
> > - Attack evals on DVS datasets raise some suspicion. In Table R6, PGD30 attacks appear to somehow perform worse than PGD10. This should not be the case, unless gradients are primarily obfuscated in a way. Did the authors question this, since this is an important flag to re-address attack evaluations (see [Carlini et al 2019], section “basic sanity tests”)?
> >
> > - On Table R7: I thought the results in Table 1 of paper compared RAT, AT and SR* models that all exploit PGD-5 examples during training (this would be the fair comparison). However in Table R7 it turns out that the “PGD5 AT” models take ~3x longer than RAT, so I guess RAT currently does not harness the same strength of adversarial training budget? If this is the case, it would have been more fair to implement RAT with PGD5 as well, and see if SR* can outperform this SOTA defense at the same cost. Perhaps authors should also clearly add their used RAT objective to the list of Eqs (3-3), (3-4),(3-5) as well (i.e., is it PGD5 or single-step examples)?

---

> > > ### Author Response · Authors · 2023-11-23
> > > **Further Response to Reviewer HCoc**
> > >
> > > **Q1. Given the results in Table R4, I believe there can be an adversary obfuscation present if one can simply manipulate the surrogate gradient parameter to attack better. Authors claim that the way attack combinations are implemented (as I did suggest) will not influence the effectiveness of their defense strategy. I disagree. Thorough evaluations are necessary in empirical defenses, and at any rate they might lead to more convincing results if defenses are reliable. More broadly, it influences the significance of the work for ML security. This work does not provide a certified robustness solution for SNNs. Since results are only empirical at CIFAR scale, the experiments and adversary evaluations should at least reveal worst-case performance (see [Carlini et al 2019], section “skepticism of results”).**
> > >
> > > We respectfully have a different viewpoint regarding your opinion. Firstly, we have followed your suggestion and conducted experiments that demonstrate the effectiveness of the SR strategy in enhancing the robustness of SNNs against ensemble attacks. Secondly, the main focus of our paper is the defense perspective rather than the attack perspective. The ensemble attacks employed in our work are more aggressive than the commonly-used single PGD attack in previous studies, which utilizes fixed surrogate and backward methods. We believe that demonstrating the effectiveness of our approach under this aggressive attack is sufficient to establish its efficacy. Thirdly, we have provided theoretical support (Theorem 1) for our algorithm, and the "certified robustness" of SNNs is beyond the scope of this paper. Additionally, we have included an experiment on the ImageNet-Tiny dataset (refer to Table R1 in the response to reviewer bv7g), which shows that models trained with SR* exhibit greater robustness compared to models trained with AT when subjected to FGSM or PGD attacks.
> > >
> > > **Q2. Differences to adv. logit-based regularization methods. Comparsion with TRADES.**
> > >
> > > We strongly believe that the differences between the SR strategy and adv. logit-based defense methods are substantial in terms of both algorithms and objectives. The SR strategy is a general defense method that directly constrain the inherent properties (gradients) of SNNs, while the adv. logit-based regularization method relies on the generated adversarial examples during the training phase. Moreover, we have performed an additional experiment on the CIFAR-10 dataset to directly compare the robustness of SNNs trained with the TRADES and SR strategies. The results are displayed in Table R9, where it is evident that the SNN trained with the SR strategy consistently outperforms the one trained with TRADES across all scenarios. This further reinforces the superiority of our approach over TRADES.
> > >
> > > **Table R9. Comparsion of SNNs trained with TRADES and SR**
> > > | Methods    | Clean | FGSM  | PGD10 | PGD30 | PGD50 |
> > > |------------|-------|-------|-------|-------|-------|
> > > | TRADES     | 90.04 | 30.25 | 11.86 |  9.28 | 8.84  |
> > > | SR         | 86.57 | 34.79 | 15.94 | 12.27 | 11.70 |
> > >
> > > **Q3. Questions about Table R7 and RAT.**
> > >
> > > In Table R7, we have adopted the settings recommended by the authors of RAT in their paper. The adversarial examples used in RAT are randomly sampled from {FGSM(BPTT), RFGSM(BPTT), FGSM(BPTR), RFGSM(BPTR)}. The authors have explicitly stated that "RFGSM is recognized as an effective substitution for PGD in adversarial learning" and one of the main objectives of RAT is to achieve energy-efficiency. Thus, we think it is appropriate to use the original settings as recommended by the authors.

---

### Official Review · Reviewer_JC8H · 2023-10-30

**Soundness:** 3 good
**Presentation:** 3 good
**Contribution:** 3 good
**Rating:** 8
**Confidence:** 4

**Summary:**

This paper proposes a framework to improve the robustness of an SNN model. They first identify the SNN robustness under random attack and adversarial attack. Then, they add a regularization term in loss to make the SNN model more robust under adversarial attack.

**Strengths:**

1.	This work analyzed the SNN robustness under random attack and adversarial attack, which provides very meaningful observations.
2.	The proposed idea is interesting that tries to shrink the gap between two attacks.
3.	Detailed experiments are presented to demonstrate the efficiency of the proposed method.

**Weaknesses:**

The specialization of the regularization term is not highlighted, see common for details.

**Questions:**

1.	In Formula 10, I think the left part tries to compute the gradient of input, however, the right parts compute the gradient of last layer.
2.	I think adding regularization in loss to improve the robustness is a widely used method. It is better to highlight the specialization, i.e. whether ANN can adopt this technique?
3.	Whether the proposed method will affect the robustness under random attack?

---

> ### Author Response · Authors · 2023-11-16
> **Response to Reviewer JC8H**
>
> Thanks for your valuable comments and suggestions. We are delight that you find our obervation meaningful, our idea interesting and experiments detailed. We would like to address your concerns and your questions in the following.
>
> **Q1: Details of Formula 10**
>
> Thanks for pointing it out. We would like to clarify the notations used in Formula (10) as follows. Let $x$ denote the image, and $\{x[1], x[2], \dots, x[T]\}$ represent the input image series. In our paper, we use $x[t]=x$ for all $t=1,\dots,T.$ The network is denoted by $f$ with parameters $w$. The output of the network $f$ is a vector $f(x,w)\in \mathbb{R}^{N \times 1}$, where $N$ represents the number of classes. The output vector $f(x,w)$ is obtained by applying the softmax function to the output of the last layer, i.e.
> \begin{equation}
>     f(x, w) = softmax \left(
>         \sum_{t=1}^T s_1^L[t], \dots, \sum_{t=1}^T s_N^L[t]
>     \right).
>     \tag{2-1}
> \end{equation}
> Therefore, the $y^{th}$ component of $f(x, w)$ where $y$ represents the true label is
> \begin{equation}
>     f_y(x, w) = softmax_y \left(
>         \sum_{t=1}^T s_1^L[t], \dots, \sum_{t=1}^T s_N^L[t]
>     \right).
>     \tag{2-2}
> \end{equation}
> According to the chain rule, the gradient of $f_y$ with respect to the input $x$ is
> \begin{equation}
>     \begin{split}
>         \nabla_x f_y(x,w) &=  \frac{\partial softmax_y }{\partial \sum_t s_1^L[t]}
>                      \cdot
>                     \nabla_x \left( \sum_t s_1^L[t] \right) + \dots  +
>                     \frac{\partial softmax_y }{\partial \sum_t s_N^L[t]}
>                      \cdot \nabla_x \left( \sum_t s_N^L[t] \right).
>     \end{split}
>     \tag{2-3}
> \end{equation}
> In Equation (2-3), the gradient of the $i^{th}$ component $\nabla_x \left( \sum_t s_i^L[t] \right)$ in the last layer with respect to $x$ can be further expresses as
> \begin{equation}
>     \nabla_x \left( \sum_t s_i^L[t] \right) =  \sum_t \nabla_x s_i^L[t]
>     = \sum_t \sum_{\tilde{t}=1}^t \nabla_{x[\tilde{t}]} s_i^L[t] .
>     \tag{2-4}
> \end{equation}
> Finally, the gradient $\nabla_x f_y(x,w)$ is written as
> \begin{equation}
>     \nabla_x f_y(x,w) = \sum_{i=1}^N \left( \frac{\partial softmax_y
>                     }{
>                         \partial \sum_t s_i^L[t]
>                     }
>                     \left(
>                         \sum_{t=1}^T \sum_{\tilde{t}=1}^t \nabla_{x[\tilde{t}]} s_i^L[t]
>                     \right) \right).
>     \tag{2-5}
> \end{equation}
>
> We revise Formula (10) in the latest version, and provide the detailed derivation of the formula in Appendix J.
>
> **Q2: Whether ANN can adopt SR?**
>
> Thanks for pointing it out! Theorem 1 also holds true for ANNs which are differentiable. Therefore, this technique can be employed to enhance the robustness of ANNs. To demonstrate this point, we conduct experiments for ANNs with VGG11 architecture on the CIFAR-10 dataset. The classification accuracy (%) is reported in Table R2-1, which demonstrates that our can also boost the robustness of ANNs.
>
> **Table R2-1. Performance of ANNs on CIFAR-10 dataset**
> | ANN_Models   | Clean | FGSM  | PGD10 | PGD30 | PGD50 |
> |--------------|-------|-------|-------|-------|-------|
> | Vanilla      | 92.91 | 6.62  | 0.00  | 0.00  | 0.00  |
> | SR (λ=0.004) | 88.64 | 48.96 | 31.39 | 27.24 | 26.77 |
>
> **Q3: Whether the proposed method will affect the robustness under random attack?**
>
> Thanks for your suggestion. To evaluate that, we report the classification accuracy (%) of models trained with SR and SR*(PGD5+SR) under random attack (eps=0.1) with WideResNet-16 on CIFAR datasets. The results are shown in Table R2-2, we can find that the SR and SR* strategy can also improve the robustness of SNNs against random attacks.
>
> **Table R2-2. Compare the random robustness of different methods on CIFAR datasets**
> |           |               | Vanilla | SR     | SR*    |
> |-----------|---------------|---------|--------|--------|
> | CIFAR-10  | clean         | 93.89   | 91.94  | 85.63  |
> |  CIFAR-10 | random attack | 67.467  | 86.327 | 81.885 |
> | CIFAR-100 | clean         | 74.59   | 67.81  | 60.37  |
> | CIFAR-100 | random attack | 26.27   | 49.9   | 48.678 |

---

> > ### Comment · Reviewer_JC8H · 2023-11-22
> > **Response to authors**
> >
> > Thanks for the detailed explanation and additional experiments, and my concerns have been solved. This work is significant as it theoretically analyzes random and adversarial attacks. Additionally, addressing the security issues in SNN is crucial, making this research valuable for both SNN security and broader AI security considerations. Based on the authors' reply, I would like to raise my score.

---

> > > ### Author Response · Authors · 2023-11-23
> > > **To Reviewer JC8H: Thanks for raising the score.**
> > >
> > > We are delighted to see that the major concerns raised by the reviewer have been successfully addressed. We would like to reiterate our deep appreciation for the reviewer's dedicated time and effort in scrutinizing our paper and providing invaluable feedback.

---

### Official Review · Reviewer_bv7g · 2023-11-02

**Soundness:** 4 excellent
**Presentation:** 4 excellent
**Contribution:** 4 excellent
**Rating:** 8
**Confidence:** 5

**Summary:**

This paper studies the adversarial robustness of spiking neural networks. First, the authors observe that SNNs exhibit robustness against random perturbations, but display vulnerability to small-scale adversarial perturbations. After that, the authors derive some theoretical results on the bounds of the gap between the robustness of SNNs under these two kinds of perturbations, and show that it is upper bounded by the sparsity of gradients of the output probability with respect to the input image. Motivated by such observations and theoretical bounds, the authors propose an algorithm to add the gradient sparsity regularization term to the loss function during SNN training to narrow the gap between these two kinds of perturbations. Various experimental results on the CIFAR-10 and CIFAR-100 datasets show that the proposed algorithm enhances the robustness of SNNs.

**Strengths:**

Originality: The related works are adequately cited. The main results in this paper will certainly help us have a better understanding of the adversarial robustness of spiking neural networks. I have checked the technique parts and found that the proofs are solid. Some strengths of this paper are listed below:
1. The authors provide several useful observations and theoretical bounds on the robustness against random perturbations and small-scale adversarial perturbations, and derive the upper bound of the gap between the robustness of SNNs under these two kinds of perturbations.

2. Based on the observations and theoretical bounds, the authors proposed a novel loss function involving the gradient sparsity regularization term, which could improve the robustness of SNNs.

3. Various results verify the effectiveness of their proposed algorithms.

Quality: This paper is technically sound.

Clarity: This paper is well written. I find it is easy to follow.

Significance: I think the results and the proposed algorithm in this paper are significant, as explained above.

**Weaknesses:**

1. The paper conducts the experiments on the CIFAR-10 and CIFAR-100 datasets. Is it possible to conduct the experiments on more large-scale datasets, such as ImageNet or ImageNet-Tiny datasets?

2. It requires some assumptions for Theorem 1 to be true, such as the function $f$ should be differentiable, and $\epsilon$ should be small enough, could you add these assumptions to Theorem 1?

Some other minor questions:
1. Line 4 in Proposition 1, Page 6,  The proof is provides -->  The proof is provided.
2. The last line in (17), Page 15,  $\epsilon$ -->  $\epsilon^2$.

**Questions:**

Please see the above weaknesses.

---

> ### Author Response · Authors · 2023-11-16
> **Response to Reviewer bv7g**
>
> Thank you for your thoughtful comments. We are encouraged that you find our paper technically sound, well-written and our results significant. We would like to address your concerns and your questions in the following.
>
> **Q1: Experiments on more large-scale datasets**
>
> We trained three VGG-11 models on the TinyImageNet dataset: a vanilla model with no robustness enhancement (Vanilla), an FGSM [eps=2/255] adversarially trained model (AT), and a model using both FGSM [eps=2/255] adversarial training and sparsity regularization (SR*). Throughout training and evaluation, we cropped all images to a resolution of 64x64, which is more complex compared to the 32x32 images from CIFAR-10 datasets.
>
> Under these conditions, the SR* model exhibited the best performance, achieving an accuracy of 10.82% under PGD50 attack and 14.89% accuracy under FGSM attack. In comparison to the pure AT model, the SR* model showed significantly better robustness against various attacks, highlighting the effectiveness of our method.
>
> **Table R1. Performance on TinyImageNet dataset**
> | ANN_Models           | Clean | FGSM  | PGD10 | PGD30 | PGD50 |
> |--------------        |-------|-------|-------|-------|-------|
> | Vanilla                       | 58.23 | 0.64  | 0.00  | 0.00  | 0.00  |
> | AT (FGSM, eps=2/255)          | 52.17 | 11.06 | 6.09  | 5.25  | 4.96  |
> | SR* (FGSM, eps=2/255 λ=0.001) | 43.44 | 14.89 | 11.49 | 10.97 | 10.82 |
>
> **Q2: Add assumptions to Theorem 1**
>
> Thanks for your suggestion! In our revised version, we have incorporated the assumptions that the function $f$ is differentiable and that $\epsilon$ is sufficiently small into Theorem 1.
>
>
> **Q3. Other minor questions.**
>
> Thanks for pointing it out! We have revised these typos and updated our paper.

---

> > ### Comment · Reviewer_bv7g · 2023-11-22
> > **Thanks the efforts of authors**
> >
> > I would like to thank the authors for the response and it has sufficiently addressed my questions.After reading the other reviews, I still believe it is a good paper that will be of interest to the SNNs community. Therefore, I would like to keep my rating.

---

> > > ### Author Response · Authors · 2023-11-23
> > > **To Reviewer bv7g: Thanks for keeping the rating.**
> > >
> > > We are delighted to see that the major concerns raised by the reviewer have been successfully addressed. We would like to reiterate our deep appreciation for the reviewer's dedicated time and effort in scrutinizing our paper and providing invaluable feedback.

---

### Author Response · Authors · 2023-11-16
**Response To All Reviewers (1/2)**

We sincerely thank all reviewers for insightful feedbacks that help us to revise our paper according to recommendations and criticisms. In this general response, we would like to address the common concerns raised by the reviewers. We have made thorough revisions to our paper, indicating all the changes by highlighting them in blue.

**Q1: Clarification of differences between the proposed SR strategy and related works [JC8H, HCoc, FfwP]**

We clarify the differences between our method, exsiting regularization method, adversarial training and its variants in this rebuttal. We have incorporated this issue into Sec. 4.3 and Appendix H in the latest version.

> The differences between our training algorithm and the work by [Finlay & Oberman](https://arxiv.org/pdf/1905.11468.pdf).
* The primary distinction lies in the fact that in SR, the regularization term is exclusively tied to the _model itself_, as opposed to being connected to the _multi-class calibrated loss function_ used during the training phase. To provide further clarification, we can express the training loss of Finlay & Oberman as  follows (defending against $\ell_\infty$ attacks):
\begin{equation}
    \text{min}_w \left( l(x,w) + \frac{\lambda}{2} \Vert \nabla_x l(x,w)\Vert_1^2 \right).
    \tag{1-1}
\end{equation}
where $l(x,w)$ is the multi-class calibrated loss function for the classification task (such as cross-entropy loss), $y$ is the label of the input $x$. In contrast, the training loss of our work is
\begin{equation}
    \text{min}_w \left( l(x,w) + \lambda \Vert \nabla_x f_y(x,w)\Vert_0 \right).
    \tag{1-2}
\end{equation}
While Finlay & Oberman' work primarily focuses on the loss function, it is important to note that the regularization term in their approach varies depending on the specific choices of $l(x,w)$. In contrast, our proposed method introduces a regularization term that remains independent of the choice of loss function. Moreover, our method can be combined with adversarial training to enhance the performance of adversarial training (refers to Table 2 of the main text).
* The objectives of these two training algorithms are different. In Finlay & Oberman's work, Equation (1-1) acts as an upper bound for the loss function in adversarial training. Their goal is to minimize the loss of adversarial training through Equation (1-1) and enhance the robustness of models. On the other hand, in our work, $\Vert \nabla_x f(x,w) \Vert_0$ represents an upper bound for the gap between the adversarial robustness and random robustness. We aim to reduce this gap and thereby improve the overall robustness of SNNs. Our objective is not directly associated with adversarial training.

> The differences between our work and (clean) logit pairing approach ([Kannan, Kurakin & Goodfellow, 2018](https://arxiv.org/abs/1803.06373)).

* The regularization terms are different. The clean logit pairing method utilizes a regularization term $\Vert f(x,w) - f(x',w)\Vert_2^2$, where $x$ and $x'$ are two randomly selected clean images. In contrast, the regularization term in our paper, $\Vert \nabla_x f_y(x,w)\Vert_0$, is theoretically related only to the input image $x$. It is important to note that the approximation formula $|\frac{f_y(x,w) - f_y(x+hd,w)}{h}|$ involves two closely situated points, namely $x$ and $x+hd$, rather than two randomly chosen images.
* The purposes of the regularization term are different. The clean logit pairing approach aims to promote similarity in probability distributions for two randomly chosen clean images. This method improves models' robustness on MNIST and SVHN, although it lacks theoretical guarantees. In contrast, the SR strategy in our paper aims to improve the sparsity of the input gradient $\nabla_x f_y(x,w)$ to reduce the gap between adversarial robustness and random robustness. The SR strategy is supported by  theoretical analysis (Theorem 1).

---

> ### Author Response · Authors · 2023-11-16
> **Response To All Reviewers (2/2)**
>
> > The differences between our work and (adversarial) logit pairing approach ([Kannan, Kurakin & Goodfellow, 2018](https://arxiv.org/abs/1803.06373)), as well as TRADES ([Zhang et al. ICML 2019](https://arxiv.org/abs/1901.08573)).
>
> * Both adversarial logit pairing approach (RFGSM-40) and TRADES (solving a maximum problem) utilize adversarial examples during their training phase. However, it is important to clarify that the proposed SR strategy does not use adversarial examples as explained in the first response to Reviewer HCoc. In other words, SR is not a kind of adversarial training.
> * The objectives are different. The primary objective of the adversarial logit pairing and TRADES is to encourage the probability distributions of the input image and its corresponding adversarial example to be similar. In contrast, our proposed SR strategy aims to induce sparsity in the gradient $\nabla_x f_y(x,w)$. We are not targeting the regularization of the probability distribution, and the regularization term corresponds to the gradient of $f_y(x,w)$, rather than the output vector $f(x,w)$.  From an intuitive perspective, minimizing the $\ell_0$ norm of $\nabla_x f_y(x)$ serves to bring $x$ closer to a local maximum point, making it harder for attackers to generate adversarial examples through gradient-based methods.

---

### Meta-Review · Area_Chair_JsWy · 2023-12-06

**Metareview:**

The paper introduces a new defence mechanism for spiking neural nets (SNNs) against adversarial attacks, using a regularization term promoting input gradient sparsity for training robust SNN. The paper is clear and well justified.

The ratings of this paper were quite diverse. One reviewer was strongly opposed to accepting the paper, mainly on the grounds that the adversarial robustness are empirical only and that the technique is rather incremental. To me the criticisms of the reviewer seem justified.

**Justification For Why Not Higher Score:**

The criticisms of the negative reviewer seem justified. The reviewer is strong and confident in their opinion, and has done substantial work in the area.

**Justification For Why Not Lower Score:**

NA

---

### Decision · Program_Chairs · 2024-01-16

Reject